# Modeling Height–Diameter Relationship for Poplar Plantations Using Combined-Optimization Multiple Hidden Layer Back Propagation Neural Network

**Jianbo Shen [1,2,3], Zongda Hu [4], Ram P. Sharma [5] 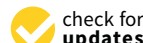, Gongming Wang [6], Xiang Meng [2,3], Mengxi Wang [2,3], Qiulai Wang [7] and Liyong Fu [2,3,*]**

[1] Academy of Agricultural Planning and Engineering, Ministry of Agriculture and Rural Affairs, Beijing 100125, China; lyshenjianbo@163.com

[2] Research Institute of Forest Resource Information Techniques, Chinese Academy of Forestry, Beijing 100091, China; 15827524590@163.com (X.M.); 15501130818@163.com (M.W.)

[3] Key Laboratory of Forest Management and Growth Modeling, National Forestry and Grassland Administration, Beijing 100091, China

[4] College of Resources, Sichuan Agricultural University, Chengdu 611130, China; huzd98@163.com

[5] Institute of Forestry, Tribhuwan University, Kritipur, Kathmandu 44600, Nepal; ramsharm1@gmail.com

[6] Institute of Biophysics, Chinese Academy of Sciences, Beijing 100101, China; gongmingwang@126.com

[7] Forestry Surveying and Designing Institute of Guangdong Province, Guangzhou 510520, China; wangqiulai2014@126.com

[*] Correspondence: fuly@ifrit.ac.cn; Tel.: +86-010-6288-9179

**Abstract:** Relationship of total height and diameter at breast height (hereafter diameter) of the trees is generally nonlinear, and therefore has complex characteristics, which can be accurately described by the height-diameter model developed using the back propagation (BP) neural network approach. The multiple hidden layered-BP neural network has several hidden layers and neurons, and is therefore considered more appropriate modeling approach compared to the single hidden layered-BP neural network approach. However, the former approach is not widely applied for tree height prediction due to absence of the effective optimization method, but it can be done using the BP neural network modeling approach. The poplar (*Populus* spp. L.) plantation data acquired from Guangdong province of China were used for evaluating the BP neural network modeling approach and compared its results with those obtained from the traditional regression modeling and mixed-effects modeling approaches. We determined the best BP neural network structure with two hidden layers and five neurons in each layer, and logistic sigmoid transfer functions. Relative to the Mitscherlich height-diameter model that had the highest fitting precision among the six traditional height-diameter models evaluated, coefficient of determination ($R^2$) of the neural network height-diameter model increased by 10.3%, root mean squares error (RMSE) and mean absolute error (MAE) decreased by 12% and 13.5%, respectively. The BP neural network height-diameter model also appeared more accurate than the mixed-effects height-diameter model. Our study proposes the method of determining the optimal numbers of hidden layers, neurons of each layer, and transfer functions in the BP neural network structure. This method can be useful for other modeling studies of similar or different types, such as tree crown modeling, height, and diameter increments modeling, and so on.

**Keywords:** Levenberg–Marquardt algorithm; *k*-fold cross-validation; traditional height-diameter functions; mixed-effects model; optimal neural network height-diameter model

## 1. Introduction

　　Tree height is one of the most important tree characteristics and measurement of which is used as a fundamental basis for evaluating forest growth and biomass, site quality, and classifying the vertical structures of a forest [1,2]. Direct measurement of tree height is generally difficult and time consuming. However, due to a strong relationship between tree height and diameter at breast height (DBH), height can be predicted using DBH as a predictor in the height-diameter model [3–12]. This method uses the measurements of tree height and DBH to fit the mathematical functions with different forms and number of parameters, and the optimal one is determined based on the standard statistical indices. This modeling approach is generally known as traditional modeling, and its main theme is to establish the mathematical equations and get the tree prediction by solving them [3–12]. However, tree height growth is substantially affected by various factors whose relationships may be in the nonlinear forms, which may pose the difficulty in describing wider variations of the tree height with a single height-diameter equation.

　　As mentioned above, generally, tree height growth has nonlinear characteristics, and is strongly correlated to various factors, such as tree size, site quality, stand density or competition and climate factors. Site factors consist of slope, altitude, soil depth, soil texture, humus layer, and soil chemical constituents. Competition is attributed to the stand crowding and density, such as number of trees, stand basal area, and canopy density. Climate factors include solar radiation, temperature, and precipitation. Because of the difficulty in acquiring the accurate information for all these factors and easy-to-apply-purpose, height prediction models are usually developed using DBH as a single predictor (simple model) or stand variables, such as basal area and number of trees per hectare are used in the models in addition to DBH (generalized model) or model incorporating DBH, stand variables and random effects (generalized mixed-effects model). In order to develop these simple or complex types of the height-diameter models, some versatile growth functions [1,9–12] are used and fitting of these functions to data using the least square regression is generally known as traditional modeling approach. However, in recent years, there has been an increasing trend of applying the mixed-effects modeling approach to account for larger variability of tree height at the subject-level (e.g., sample plot level) and increase the model's prediction accuracy [4,6,7].

　　The back propagation (BP) neural network is one of the machine learning methods, which is a multiple layer feed-forward network trained by an error inverse propagation algorithm. The BP neural network is a modern modeling approach and can be used to develop various forest models. The BP neural network is often composed of the input layer, hidden layer and output layer. The BP neural networks can realize the mapping function from input to output, and can approximate any nonlinear continuous function with high precisions. The BP neural network is characterized with transfer functions that can be selected between the input layer and hidden layers, and hidden layers have different functions, such as logistic sigmoid and tangent sigmoid functions. The transfer functions selected between the layers are also different, and therefore their expressions are different. Thus, the neural network properly represents the various forms of nonlinear effects. In recent years, the neural network is increasingly applied for predicting forest dynamics with precise results. The neural network modeling approach was used to predict different stand and individual tree characteristics, such as height [13], diameter distribution [14,15] and stem volume [16], and to establish the models of height-diameter relationships [17], growth and yield [18], inside-bark diameter and heartwood relationships [19], and to assess forest biomass [20]. These studies compared the fitting precisions of the traditional regression models of different forms with the neural network models and showed higher precisions of the neural network models. Özçelik et al. [17] and Castaño-Santamaría et al. [13] compared the mixed-effects models with neural network and their results showed higher precisions of the mixed-effects models than those of the neural network models. All these studies were based on a single hidden layer neural network, and therefore their analyses lack sufficient performance analyses and comparisons of the multiple hidden layered-neural networks. Furthermore, none of the previously applied neural network modeling approach has proposed the methods for determining the

optimal structure of neural network (determining optimal number of hidden layers and number of neurons in each hidden layer of neural network).

Since tree height-diameter relationship is substantially affected by various factors that may be nonlinearly related, the traditional height-diameter equation cannot accurately simulate growth and development of the tree height. This method has other shortcomings, such as low fitting precision and complex operational steps associated with the fitting procedures. However, the BP neural network modeling approach has both the higher fitting efficiency and higher precision, and therefore has a great usefulness in the forest modeling researches. However, current application of the neural network in the tree height-diameter modeling is limited to a single hidden layer neural network, which lacks the sufficient performance analyses and comparison of the differences in optimizing the neural network structure. This study thus intends to solve this problem, mainly improving the performance of the neural network structure through height-diameter modeling.

Using the poplar tree height and DBH data collected from Guangdong Province in China, this study establishes the multiple hidden layered-BP neural network height-diameter model and analyzes the difference of a single hidden layered- and multiple hidden layered-BP neural network modeling approaches using the MatLab 2016b software (MathWorks, Natick, MA, USA). This study also compares the fitting precision of the optimal BP neural network height-diameter model with that of the traditional regression height-diameter models and mixed-effects height-diameter models. The presented result will be important basis for developing height-diameter models using the BP neural network and predicting tree height. The proposed methods can be useful for other modeling studies of similar or different types, such as tree crown modeling, height and diameter increment modeling, and so on.

## 2. Materials and Methods

### 2.1. Data Materials

The data we used came from the sample plots that were established on the poplar plantations in the Guangdong province of China to develop the height-diameter models. The square-shaped sample plots with an area of 666.67 $m^2$ were established in the plantations. We only used the sample plots with a stand density of more than 300 trees per hectare and normal records in the tree height. A total of 9659 trees in 112 sample plots (of which 20 sample plots were measured in 1997 and 92 sample plots in 2002) were utilized for modeling height-diameter relationship. We calculated the means of height (hereafter height) and means of DBH (hereafter DBH) by sample plots for easy-to-fit purpose, especially for neural network fitting. We divided the sample plots randomly into two parts: one for training the model (80 sample plots, also defined as a fitting data set) and another for testing the model (32 sample plots, also defined as a validation data set) by application of the k-fold cross-validation method. Summary statistics of both fitting and validation datasets are presented in Table 1 and scattered graph of tree height against DBH is presented in Appendix A (Figure A1).

**Table 1.** Summary of the statistics for the model fitting and validation datasets (Min, minimum; Max, maximum; Mean, average value; Std = standard deviation).

| Data | Variable | Min | Max | Mean | Std |
|---|---|---|---|---|---|
| Fitting data set | DBH (cm) | 5.9 | 25.7 | 11.5 | 3.0 |
| | Height (m) | 2.5 | 13.0 | 8.3 | 1.9 |
| Validation data set | DBH (cm) | 7.0 | 22.0 | 10.4 | 3.4 |
| | Height (m) | 3.7 | 14.1 | 7.5 | 2.2 |

### 2.2. Modelling Approach

We developed height-diameter models using three different modeling approaches: traditional least squares regression, mixed-effects modeling, and artificial neural network approach. We focused

more on modeling height-diameter relationship using the last approach, for example, BP neural network. The optimal BP neural network height-diameter model obtained from several alternative models was compared against the height-diameter models fitted using the traditional regression and mixed-effects modeling approaches.

### 2.2.1. Traditional Approach

This involves fitting of the traditional height-diameter functions using ordinary least square regression implemented by the nls function in R software (version 3.2.2) based on fitting data set [21]. We considered six commonly used versatile height-diameter equations (Table 2) for the purpose. Since all these are the power exponential equations, they are more complex in fitting compared to other forms of the equations (e.g., linear and fractional forms) and choosing the best performing one would be more difficult also.

**Table 2.** Traditional height-diameter equations (H, height (m); DBH, diameter at breast height (cm); a, b, and c are parameters to be estimated).

| Name of Equation | Form of Equation | Source |
|:---:|:---:|:---:|
| Richards | $H = 1.3 + a(1 - \exp(-bDBH))^c$ | [9] |
| Logistic | $H = 1.3 + a/(1 + b\exp(-cDBH))$ | [22] |
| Gompertz | $H = 1.3 + a\exp(-b\exp(-cDBH))$ | [23] |
| Korf | $H = 1.3 + a\exp(-bDBH^{(-c)})$ | [24] |
| Mitscherlich | $H = 1.3 + a(1 - \exp(-bDBH))$ | [25] |
| Schumacher | $H = 1.3 + a\exp(-b/DBH)$ | [26] |

### 2.2.2. Mixed-Effects Modeling Approach

We considered the sample plot-level effect as a random effect to establish the mixed-effects height-diameter model. In order to get the convergence with the global minimum, a relatively less complex function (Schumacher function, Table 2) was chosen to include the random effect. Our main intention of developing mixed-effects height-diameter model was to compare its performance against the model obtained from the BP neural network modeling approach.

We evaluated three different variance-stabilizing functions (exponential function, power function and power function with constant) for their effectiveness in removing the heteroskedasticity problem. Akaike's information criterion (AIC), Bayesian information criterion (BIC), and log likelihood (logLik) criteria were used to select the most effective variance-stabilizing function.

The parameters in the developed mixed-effects height-diameter model were estimated by maximum likelihood using the Lindstrom and Bates (LB) algorithm implemented in the R software (version 3.2.2) nlme function based on fitting dataset [27]. Detailed descriptions of the mixed-effects modeling are presented in the references [28–31].

### 2.2.3. BP Neural Network

As pointed out in the introduction section, artificial neural network has been increasingly applied to forest growth and yield modeling in recent years [13–20]. It has the tremendous advantages on nonlinear mapping, adaptive generalization and fault tolerance, which can make up of the shortcomings of traditional modeling approaches. However, most of the existing modeling studies are based on the single hidden layer neural network, and application of the multiple hidden- layered-neural network, e.g., BP neural network in forestry has been rarely reported. The main reason for this is due to the absence of optimization method used to predict the hidden layer numbers, the number of nodes and the transfer functions of the neural network. The neural network modeling studies have shown that more the complex problem, the higher would be usefulness of the multiple hidden layers [32]. The BP neural network is suitable for function approximation, pattern recognition and classification [33]. Considering the above-mentioned advantages of the BP neural network,

we developed the neural network height-diameter models in this study, which were expected to be more accurate than those obtained from traditional regression and mixed-effects modeling approaches. The structural parameters of the BP neural network include the number of hidden layers, number of nodes in each layer, and transfer functions between the layers [34].

- Setting up of the BP Neural Network Structure

We established the tree height-diameter model based on the multiple hidden-layered BP neural network. For this, firstly, we set the range and step size of the hidden layers, number of nodes and transfer functions, and secondly, the values in a reasonable range were obtained, so as to generate a series of neural network height-diameter models. Finally, we used the k-fold cross-validation to identify the best performing one among several height-diameter alternatives.

The number of nodes in each hidden layer needs to be determined according to fitting precision. Equation (1) represents a commonly used determination method [35].

$$S = \sqrt{n + o} + m \tag{1}$$

where $S$ is the number of hidden layer nodes, $n$ is the number of nodes in the input layer, and $o$ is the number of nodes in the output layer, and $m$ is an integer ($m = 1, 2, \ldots, 10$).

The transfer functions, which occur between the hidden layers or between the input layer and hidden layer, are S-shaped logistic sigmoid and tangent sigmoid functions. The former is a unipolar S-function and the latter is a bipolar S-function. The expressions of logistic sigmoid function and tangent sigmoid function are represented by Equation (2) and Equation (3), respectively. The transfer function occurring between the hidden layer and output layer is a linear function, and its expression is represented by Equation (4):

$$f(x) = 1/(1 + e^{-x}) \tag{2}$$

$$f(x) = 2/(1 + e^{-2x}) - 1 \tag{3}$$

$$f(x) = ax + b \tag{4}$$

In order to get the best BP neural network structure for a given data, we set the selection range of the number of hidden layers, the number of nodes in each layer and the transfer function, and then generated several height-diameter neural network models. Based on the mean squared error and the number of iterations, the best performing model was identified. The exhaustive analyses of all the network models would not be appropriate. Thus, we applied the "trial and error approach" [36] to reduce the number of tests and applied the k-fold cross-validation [37] to improve the test results of the BP neural network structure.

- Normalizing Input and Output Factors

As the input factors may have different measurement units, the existence of singular samples in data would cause an increased network training time and it may also lead to the non-convergence of neural network. In response to this problem, we used the mapminmax function to normalize the input and output factors and mapped them to a scale between −1 and 1. The anti-normalization approach was used to program the results of the operation from the interval [−1,1] mapped to an actual prediction.

- Training the Model

We trained the BP neural network applying the Levenberg-Marquatdt (L-M) algorithm [38,39]. This algorithm does not follow a single negative gradient direction for each iteration, but allows the errors to be searched in the direction of deteriorating. At the same time, through the adaptively adjustment of the steepest gradient descent method and the Gaussian–Newton method optimizes the

network weight, so that the neural network can effectively converge. Equation (5) is used for adjusting the weights and thresholds.

$$\Delta w = -(J^T J + \mu I)^{-1} J^T e \tag{5}$$

where $\Delta w$ is the adjusted weights and thresholds, $I$ is unit matrix, $J$ is the Jacobian matrix of the error-weight differential, and $e$ is the vector of errors, $\mu$ is an adaptively adjusted scalar that increases as it approaches the steepest descent method with small learning rate, and when it descends to 0, the algorithm becomes a smooth harmonic between the Gauss-Newton methods.

While training the model, the parameters were set as follows: learning rate 0.01, maximum number of iterations 1000, target precision 0.001, maximum number of verification failures 20, and minimum performance gradient 0.000001.

- Model Evaluation

We used the coefficient of determination ($R^2$), root mean square error (RMSE) and mean absolute error (MAE) as evaluation indices to compare the models based on validation data set. These indices were calculated using formulae (6), (7), and (8), respectively.

$$R^2 = 1 - \sum_{i=1}^{n} \frac{(Y_i - \hat{Y}_i)^2}{(Y_i - \overline{Y})^2} \tag{6}$$

$$RMSE = \sqrt{\frac{1}{n} \sum_{i=1}^{n} (Y_i - \hat{Y}_i)^2} \tag{7}$$

$$MAE = \frac{\sum\limits_{i=1}^{n} |Y_i - \hat{Y}_i|}{n} \tag{8}$$

where $n$ is the number of samples; $\overline{Y}$, $Y_i$, and $\hat{Y}$ are mean value, measured value and predicted value of a response variable in the model (tree height, in our case), respectively.

Theoretically, the closer the determination coefficient to 1, the smaller the root mean square error and the mean absolute error, and the higher would be the model's fitting precision.

- Model Selection

We first obtained a series of the BP neural network height-diameter models by setting different values of the structural parameters. Then after, the k-fold cross-validation method was used to select the most suitable model [37]. When there is a sample set $S$ containing $m$ data records, and $t$ models to be chosen are $M_1$, $M_2$, ... , $M_t$, k-fold ($k = 5$) cross-validation procedures would be as follow:

Step 1. A sample set $S$ is randomly divided into $k$ disjoint subsets, the number of samples in each subset is $m/k$, and these subsets are denoted by $S_1$, $S_2$, ... ... , $S_k$.

Step 2. For each model $M_j(j = 1, 2, \cdots, t)$, following is done: For $n = 1$ to $k$ { Take $S_1 \cup \cdots \cup S_{n-1} \cup S_{n+1} \cup \cdots \cup S_k$ as a training set; Train the model $M_j$, and get the corresponding hypothetical function $H_{jn}$; Take $S_n$ as a verification set, and calculate the model $M_j$ generalization error $\varepsilon_{S_n}(H_{jn})$. } Calculate the average of $\varepsilon_S(H_{jn})$, $n = 1, 2, \cdots, k$, and get the average generalization error of model $M_j$.

Step 3. Calculate the average generalization error of all the models, and select the model $M_p$ with the smallest average generalization error, which is the best model.

It is noted that, as Arlot and Lerasle [40] recommended, five-fold ($k = 5$) cross-validation was applied in this study.

In general, the mean square errors are used to represent the generalized errors $\varepsilon_{S_n}(H_{jn})$, as shown in Equation (9).

$$MSE = \sum_{i=1}^{n}(Y_i - \hat{Y}_i)^2 / n \qquad (9)$$

where, $n$ is the number of samples, $Y_i$, $\hat{Y}_i$, respectively, for all observed height values and model predicted height values.

In addition to the generalized error $\varepsilon_{S_n}(H_{jn})$, the number of iterations, running time, and other criteria are also used to select the best performing model.

Tree height-diameter modeling process using the BP neural networks is shown in Figure 1. According to data situation and actual demand, we first set the implied layer number, the number of hidden layer and the number of nodes, range of values of the transfer function. Then, after we used the "trial and error approach" to determine the actual value of these structural parameters and generate the $N$ number of height-diameter models. Finally, the optimum BP neural network height-diameter model was selected through the k-fold cross-validation.

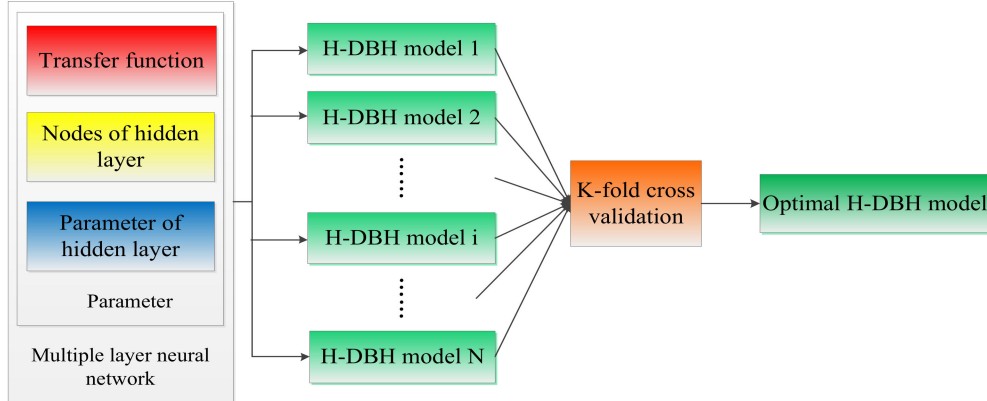

**Figure 1.** Flow chart for selecting an optimal tree height-diameter model.

The computations including the developed BP neural network height-diameter models and 5-fold cross-validation were implemented in the MatLab 2016b software (MathWorks, Natick, MA, USA).

## 3. Results

### 3.1. Model Generation and Performance

We used MatLab 2016b (MathWorks, Natick, MA, USA) to build the neural network height-diameter models through writing M program. Several neural network height-diameter model alternatives were generated when we set the number of hidden layers from 1 to 3, the number of neurons in each hidden layer from 2 to 11 with the step size of 3, the neurons adapted through "trial and error approach", and logistic sigmoid and tangent sigmoid transfer functions employed with DBH and height as input layer and output layer. The model with the smallest RMSE and MAE and the largest $R^2$ was then identified using the k-fold cross-validation with $k = 5$. The iterative results are listed in Tables 3 and 4, and more results are in Appendix A (Table A1).

**Table 3.** Performance of neural network with single hidden layer, MSE: mean square errors, logsig: logistic sigmoid function, tansig: Tangent sigmoid function.

| Neurons in Each Layer | MSE | Iterations | MSE | Iterations |
|---|---|---|---|---|
| | logsig | | tansig | |
| 1: 2: 1 | 0.1107 | 24.0 | 0.3057 | 16.0 |
| 1: 5: 1 | 0.0827 | 14.3 | 0.0669 | 15.0 |
| 1: 8: 1 | 0.0568 | 13.1 | 0.1143 | 17.0 |
| 1:11: 1 | 0.0556 | 11.0 | 0.0437 | 13.1 |

**Table 4.** Performance of neural network with two hidden layers, MSE: Mean square errors.

| Neurons of Each Layer | MSE | Iterations | MSE | Iterations | MSE | Iterations | MSE | Iterations |
|---|---|---|---|---|---|---|---|---|
| | log:log | | tan:tan | | tan:log | | log:tan | |
| 1:2:2:1 | 0.1237 | 15.2 | 0.0881 | 39.6 | 0.0790 | 22.6 | 0.1071 | 16.6 |
| 1:2:5:1 | 0.0764 | 15.2 | 0.1751 | 19.4 | 0.0446 | 21.4 | 0.1087 | 23.0 |
| 1:2:8:1 | 0.0743 | 17.6 | 0.0813 | 16.6 | 0.0884 | 23.4 | 0.1467 | 18.6 |
| 1:2:11:1 | 0.0839 | 19.8 | 0.0935 | 21.6 | 0.0801 | 15.8 | 0.0948 | 20.6 |
| 1:5:2:1 | 0.0626 | 17.0 | 0.0936 | 13.0 | 0.0558 | 14.2 | 0.0757 | 15.0 |
| 1:5:5:1 | 0.0416 | 16.2 | 0.0854 | 19.4 | 0.1017 | 15.0 | 0.0832 | 22.4 |
| 1:5:8:1 | 0.0897 | 14.2 | 0.0584 | 17.2 | 0.1091 | 16.2 | 0.0971 | 12.8 |
| 1:5:11:1 | 0.0764 | 17.6 | 0.1323 | 20.6 | 0.0950 | 14.4 | 0.1214 | 12.0 |
| 1:8:2:1 | 0.0610 | 14.6 | 0.0986 | 17.0 | 0.0644 | 13.8 | 0.1582 | 14.4 |
| 1:8:5:1 | 0.0971 | 13.6 | 0.0948 | 13.4 | 0.0869 | 14.0 | 0.1304 | 18.8 |
| 1:8:8:1 | 0.0599 | 21.0 | 0.0841 | 12.2 | 0.0978 | 13.4 | 0.0586 | 13.2 |
| 1:8:11:1 | 0.1060 | 13.6 | 0.1441 | 17.4 | 0.1004 | 13.0 | 0.0983 | 13.4 |
| 1:11:2:1 | 0.0750 | 22.4 | 0.0852 | 17.8 | 0.0530 | 15.0 | 0.1272 | 16.4 |
| 1:11:5:1 | 0.1573 | 12.8 | 0.2539 | 17.6 | 0.1400 | 12.2 | 0.0887 | 18.8 |
| 1:11:8:1 | 0.1058 | 23.4 | 0.1720 | 17.0 | 0.1020 | 15.8 | 0.0917 | 12.2 |
| 1:11:11:1 | 0.1402 | 12.8 | 0.2726 | 23.4 | 0.0766 | 14.4 | 0.1651 | 13.0 |

The MSE and the number of iterations were used as evaluation indices in screening the models and the performance statistics of different numbers of the hidden layers corresponding to the neural networks are presented in Table 5. When the number of hidden layer was 1, there were 8-networks structure combinations. When the number of hidden layers was 2, there were 64 network structure combinations. When the number of hidden layers was 3, there were 512 network structure combinations. Difference of the average MSE of the double hidden layers from that of the triple hidden layers was not substantially large even though the neural network with the double hidden layers, the structure of which is 1:5:5:1, had the smallest MSE. There was an indication that this structure, which provided the best precision, could be used as an optimal neural network structure of the height-diameter model.

There was a slight difference between the double-hidden layer and single hidden layer in the number of iterations, but it was much lower than the three-hidden layer, which showed that the convergence rate of the double-hidden layer and single hidden layer were almost similar but slightly higher than the three-hidden layers. Taken together, double hidden layer had a higher precision. We selected the double hidden layers (Table 4). When the number of neurons in each layer was 1:5:5:1, and the logistic sigmoid functions were all selected, the minimum value of MSE was 0.0416. The neural network height-diameter models generated with structure had the best fitting performance. The number of iterations required to generate the best neural network model was not necessarily the minimum number of iterations corresponding to the hidden layer (Table 5). The reason for this is that the number of iterations required to obtain the minimum increase of MSE.

**Table 5.** Statistics for performance of three neural networks, MSE: Mean square errors.

| Category | Number of Neural Network | Average MSE | MSE (min) | Ratio of MSE < 0.1 | Average Iterations | Iterations (min) | Best Neural Network Structure | | |
|---|---|---|---|---|---|---|---|---|---|
| | | | | | | | Neurons of Each Layer | Transfer Function | Iterations |
| Single hidden layer | 8 | 0.1045 | 0.0437 | 62.50% | 15 | 11.0 | 1:11:1 | tansig | 13.0 |
| Double hidden layer | 64 | 0.1027 | 0.0416 | 62.50% | 16 | 12.0 | 1:5:5:1 | logsig:logsig | 16.2 |
| Triple hidden layer | 512 | 0.1010 | 0.0428 | 62.89% | 21 | 11.6 | 1:8:8:8:1 | tansig:tansig:logsig | 16.6 |

### 3.2. Transfer Function

We generated several neural network models (Figure 2) and selected the best performing one. This figure shows that the number of neurons of both the first hidden layer and second hidden layer were 5; the first hidden layer neurons were $H_{11}$, $H_{12}$, $H_{13}$, $H_{14}$, $H_{15}$, the second hidden layer neurons were $H_{21}$, $H_{22}$, $H_{23}$, $H_{24}$, $H_{25}$. Different layers would have different biases, namely $Bias_1$, $Bias_2$, $Bias_3$, respectively. The transfer functions of the input layer and the first hidden layer was the logistic sigmoid function, the transfer function of the first hidden layer and the second hidden layer was also the logistic sigmoid function, and the output layer was the purelin function.

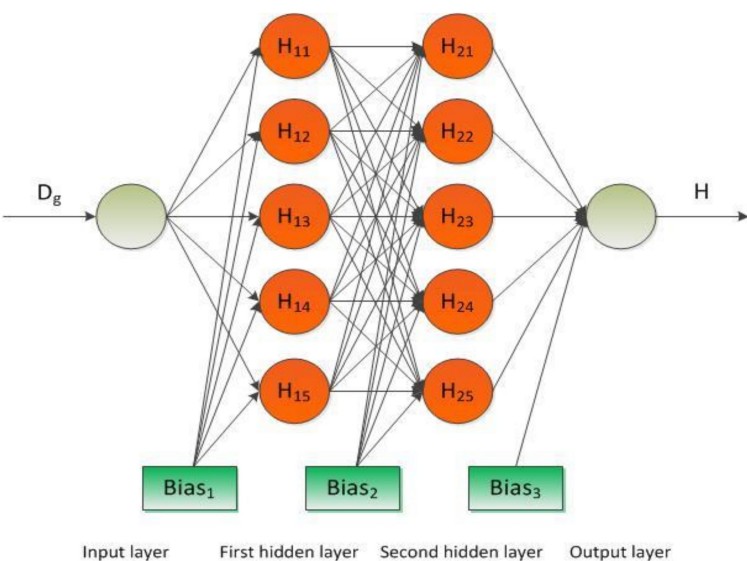

**Figure 2.** Height-diameter model of BP neural networks.

### 3.3. Comparison with Traditional Model Accuracy

We compared the height-diameter models (Table 2) fitted using ordinary least squares regression and BP neural network height-diameter model based on three evaluation indices (Table 6). Parameter estimates of all the traditional height-diameter models were significant ($p < 0.05$) and they are presented in the Appendix A (Table A2).

**Table 6.** Evaluation indices of the BP neural network height-diameter models and traditional height-diameter model ($R^2$, coefficient of determination; RMSE, root mean square error; MAE, mean absolute error).

| Evaluation Index | Richards | Logistic | Gompertz | Korf | Mitscherlich | Schumacher | Neural Network |
|---|---|---|---|---|---|---|---|
| $R^2$ | 0.6794 | 0.6774 | 0.6771 | 0.6757 | 0.6837 | 0.6788 | 0.7541 |
| RMSE | 1.2713 | 1.2752 | 1.2758 | 1.2786 | 1.2628 | 1.2724 | 1.1133 |
| MAE | 1.0887 | 1.1041 | 1.0972 | 1.0858 | 1.0963 | 1.0809 | 0.9482 |

The BP neural network modeling approach appeared better than the traditional regression approach for establishing the tree height-diameter models. Among the traditional models evaluated (Table 2), the highest fitting precision was found with the Mitscherlich model, and the lowest fitting precision was with the Logistic model. The BP neural network height-diameter model had larger $R^2$ (by 10.3%), and smaller RMSE (by 12%) and MAE (by 13.51%) than those of the Mitscherlich height-diameter model. The tree heights predicted from the best BP neural network height-diameter model and the Mitscherlich height-diameter model were compared against the observed height (Figure 3). The prediction accuracy of the former model shows a substantially higher accuracy than the latter model.

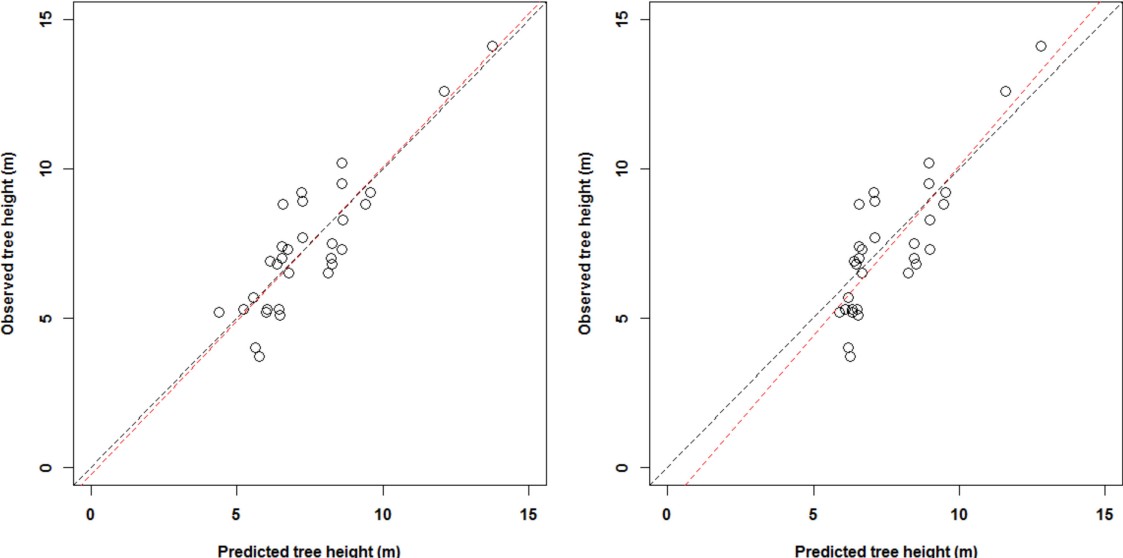

**Figure 3.** Comparison of height prediction by two different modeling approaches: left graph is based on the back propagation (BP) neural network approach, and right is based on the traditional approach (Mitscherlich model). Fitting equation based on the observed data and predicted data, y = 1.0927x − 0.2554 for left graph, and y = 1.1433x − 1.306 for right graph.

The residuals for the BP neural network model are concentrated around 0 (Figure 4), indicating that this model has better fitting performance. Using the two intervals of (−1,1) and (−2,2) in which our data points scattered mostly as examples, Figure 4 shows that using the BP neural networks height-diameter model, 20 points of the residual are within the range of (−1,1), accounting for 62.5%, and range of (−2,2) has 30 points, accounting for 93.75%. However, using the Mitscherlich height-diameter model, 15 points are within the range of (−1,1), accounting for 50%, and range of (−2,2) has 28 points, accounting for 87.5%. Thus, the precision of the BP neural network model appeared higher than the Mitscherlich model.

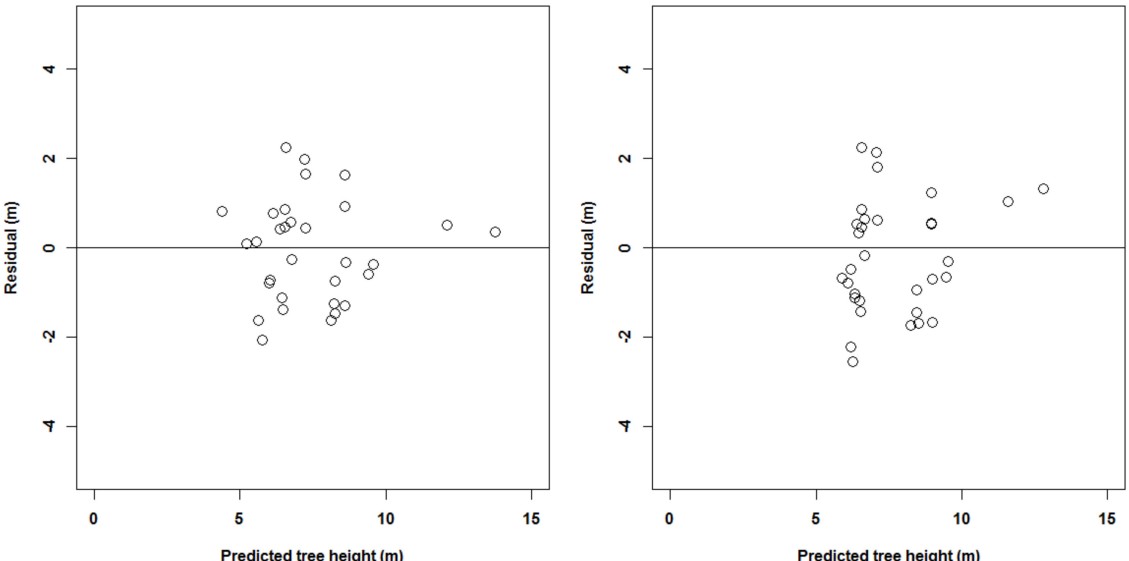

**Figure 4.** Residual graphs: left is based on the BP neural network height-diameter model and right is based on the Mitscherlich height-diameter model.

### 3.4. Comparison with Mixed-Effects Model Accuracy

We included random effects at sample plot-level into the simplest traditional height-diameter model (Schumacher model) among the six models presented in Table 2. The random effect added to the parameter *b* of this model converged with the smallest AIC and BIC, and the largest log likelihood among the alternative mixed-effects models formulated through addition of the random effect to the fixed-effect parameters. Exponential function showed the most powerful ability to account for the heteroskedasticity and thus was used to develop the final mixed-effects height-diameter model (Table 7).

**Table 7.** Evaluation indices of three variance-stabilizing functions employed to the nonlinear mixed-effects height-diameter model; AIC, Akaike information criterion; BIC, Bayesian information criterion; loglik, log likelihood.

| Variance Function | Formula | AIC | BIC | loglik |
|---|---|---|---|---|
| Exponential function | $\mathrm{var}(\varepsilon) = \sigma^2 \exp(\beta DBH)$ | 283.8982 | 295.8084 | −136.9491 |
| Power function | $\mathrm{var}(\varepsilon) = \sigma^2 DBH^{\beta}$ | 283.9949 | 295.9051 | −136.9975 |
| Power function with constant | $\mathrm{var}(\varepsilon) = \sigma^2 (\beta_1 + DBH^{\beta_2})^2$ | 285.9949 | 300.2871 | −136.9975 |

The mixed-effects height-diameter models with estimated parameters are presented in Equation (10).

$$
\begin{cases}
H_i = 1.3 + 17.8522 \exp((-10.5868 + u_i)/DBH_i) + \varepsilon_i \\
u_i \sim N(0, \psi = 0.3966) \\
\varepsilon_i \sim N(0, R_i = 8.4698 \exp(-0.1422 DBH_i))
\end{cases}
\tag{10}
$$

where $H_i$ and $DBH_i$ are the stand mean height and stand mean diameter at breast height of the $i$th sample plot, $u_i$ is the random effects generated by the $i$th sample plot and assumed to be distributed normally with zero expectation and a variance-covariance matrix $\psi$, and $\varepsilon_i$ is the error term of $i$th sample plot and also assumed to be distributed normally with zero expectation and a variance-covariance matrix $R_i$.

The fitting precision of the mixed effects height-diameter model ($R^2 = 0.7179$, RMSE = 1.1926, and MAE = 0.9888) was substantially lower than that of the neural network height-diameter model, but higher than that of the traditional height-diameter model (Equation 10, Table 6). We considered that when a mixed effects model was used, it would be equivalent to adding more input factors. Then, it was necessary to change the structure of the neural networks and added random effect factor as the input to the neural network for comparison purpose, that is, it would be more meaningful to compare the same number of input factors.

## 4. Discussion

We established the modeling method that could generate several BP neural network height-diameter models based on the combinatorics mathematics. Among these models, we selected the best model through the comparison of the fitting and prediction accuracies, and convergence rates. The BP neural network optimization method was employed to establish the optimal height-diameter model for poplar plantations in the Guangdong Province in China. The poplar tree species has been becoming one of the main plantation tree species in China in recent decades due to its faster growing characteristics, and is recognized as a focus for research of the woody plants and ideal materials for bioenergy research. Also, it is of great importance for taking poplar as a research objective in our study. This study compares the fitting performances of a single hidden layered- and multiple hidden layered-BP neural network approaches. When there were two hidden layers, the higher performance was obtained, i.e., the neural network with double hidden layers is attributed to a higher fitting precision, higher estimating efficiency, better acceptable time of the iterations. This is because that, increasing the number of hidden layers may not only result in a longer computational time, but also increases the likelihood of over fitting, which results in the model's non-optimal prediction performance. In the past, most studies focused on single hidden layer neural networks [13–20], but none of them investigated the effects due to more numbers of hidden layers, neurons and transfer functions of the neural networks. In this context, our study, which focused on these features of the neural network modeling, may be interesting and useful to other researchers.

The traditional Mitscherlich height-diameter model with the highest fitting precision was compared with the optimized multiple hidden layered-BP neural network height-diameter model. The BP neural network model appears substantially superior to the Mitscherlich model in predicting tree height (Table 6, Figures 3 and 4). Our results are also consistent with those from Castaño-Santamaría et al. [13] and Özçelik et al. [17], which predict tree height of the uneven-aged beech forests in northwestern Spain and Crimean juniper in southwestern region of Turkey. These studies compared the neural network models against the nonlinear regression models. Although our study is based on the different tree species from those studied by Castaño-Santamaría et al. [13] and Özçelik et al. [17], comparison results are almost identical, meaning that neural networks can be the best alternative of modeling on any tree data regardless of species. Özçelik et al. [17] used the single hidden layer with only one or two hidden nodes in the neural network and they did not investigate the effects of multiple hidden layers on the precision of the neural network model and determination of appropriate forms of the transfer functions. Our study is substantially different from the previous studies [13–20], because we proposed the method

of selecting the optimal model through application of the "trial and error approach" [30], k-fold cross-validation approach [31] and combinatorial optimization approach. It can help determining the structure of the neural network, such as the hidden layer nodes, transfer functions and the number of hidden layers. The hidden layers, number of neurons in each layer, and transfer functions can have substantial effects on the precision of the neural network model. Castaño-Santamaría [13] considered the change of input factors, but did not take into account other factors, such as different transfer functions of the neural network, and did not introduce the process of determining the optimal neural network model. Castro et al. [18] established the multi-layer perceptron neural network growth model for Eucalyptus. They estimated the annual mortality with the best structure associated with three neurons in the input layer, four neurons in the hidden layer, and one neuron in the output layer. All these studies [13,17,18] compared the precisions using different input variables, but none of them compared different numbers of transfer functions and hidden layers, and neurons of neural networks.

In our study, the neural network modeling produced the highest fitting precision and prediction accuracy, followed by mixed-effects models, and finally non-linear traditional regression models (Table 6, Equation (10)). This might be related to our employed methods of the structural optimization of the BP neural network. Fitting precisions of three different modeling approaches, such as traditional regression, mixed-effects modeling and neural network modeling were also compared in the previous modeling studies [13,17]. These studies showed the highest precisions of mixed-effects models, followed by neural network models, and traditional regression models. This may be due to the weaknesses of their modeling techniques, whereby they did not investigate the effects of multiple hidden layers on the precisions of the neural network models they developed. The results obtained from different modeling approaches including ours thus indicate the inconsistent ranking of approaches on the basis their fitting and prediction accuracies. Further investigation on the models, especially those to be developed with mixed-effects modeling and neural network modeling approaches using large datasets collected from extensive forest areas is necessary to better explore their differences. Generally, the neural network models have strong robustness, but traditional regression models and mixed-effects models have biological significance, for example, they have the parameters describing growth rates and growth patterns. The neural network modeling approach has tremendous advantages, such as avoiding complex selection procedures for the best performing model and obtaining higher precision. Furthermore, the BP neural network modeling has a better generalization ability, and therefore can approximate any nonlinear continuous function with a high precision. The BP neural network modeling approach is suitable for describing plant growth, which generally follows nonlinear patterns, and making it suitable for solving the problems caused by interrelated factors affecting plant growth.

The process of determining the selection of the traditional height-diameter models and neural network height-diameter models was also evaluated in this study. The traditional regression modeling needs the evaluation and comparison of differences among the fitting precisions of the candidate models considered, and the fitted model with the highest precision could be selected as the final model. The neural network modeling, on the other hand, needs the determination of the number of hidden layers and the number of neurons in each layer, and the numbers and forms of the transfer functions, and this modeling approach determines the best structure of the BP neural network. There are none of the well-organized robust methods, which can determine the number of hidden layers and the number of transfer functions, and the number of neurons based on the combinatorial mathematics that could help obtain the high precisions. We applied this method, and thus the height-diameter model proposed in this article is based on the multiple hidden layered-BP neural network. This method can help modelers to find the best neural network structure, and thus provide the best performance. However, neural network weights and thresholds are not easy to explain and determining the optimal structure of neural network is more tedious, and this can be done into a friendly interface program module, which can help future modelers to quickly determine the best structure of the BP neural network.

Our modeling system introduces the method of determining the best neural network structure with optimal numbers of hidden layers and neurons in each layer, and optimal number and forms

of transfer functions. The model comparison would make the senses when traditional regression models, mixed-effects models, and neural network models have the same input factor, such as DBH in our case. Because DBH is the main factor influencing tree height, in this study, only the effect of DBH on the tree height was considered for both modeling approaches. In our subsequent studies, in addition to DBH, the effects of other factors on the tree height growth may be considered and models can be made more comprehensive and complex. However, introducing many variables does not guarantee the high accuracies of the models that are developed using any modeling approach. In the neural network modeling, all potential interconnected factors that affect the response variable (tree height, in our case) of the model are assumed to be properly described, and thus this approach can be considered more appropriate than other modeling approaches, such as ordinary least square regression and mixed-effects modeling approaches. This is because

The neural network is able to optimize the model efficiently through the combinatorial optimization process.

## 5. Conclusions

We proposed the modeling method which can generate several BP neural network models based on the combinatorics mathematics, and among them, the model with the best structure was selected through comparison of the fitting and prediction precisions, and convergence rate. In the process of determining the structure of the neural network, both the number of hidden layers, the numbers of neurons and the number of transfer functions of the BP neural network were considered. We developed the BP neural network optimization method to establish the optimal tree height-diameter model for poplar plantations in the Guangdong Province of China. The optimal BP neural network structure was 1:5:5:1 and transfer functions determined were the logistic sigmoid functions. The optimal structure of the BP neural network height-diameter accounted for 75% variations of tree height-diameter relationship, which is higher (by 10.3%) than the best fitted traditional regression height-diameter model. The BP neural network height-diameter model also outperformed the mixed-effects height-diameter model. In addition to diameter at breast height, tree height growth is also substantially affected by other several factors, such as site and climate factors, and stand conditions, which may be introduced in the height-diameter model for gaining a higher prediction accuracy. The proposed method of the neural network modeling can be suitable for other forest modeling studies of similar or different types, such as tree crown modeling, height and diameter increments modeling, and so on.

**Author Contributions:** J.S., G.W. and R.P.S. conceived the idea and designed methodology; J.S., L.F., X.M. and M.W. collected data; J.S., Z.H., G.W., L.F. and Q.W. analyzed data; J.S., L.F., R.P.S. and G.W. wrote manuscript and contributed critically to improve the manuscript, and gave a final approval for publication. All authors have read and agreed to the published version of the manuscript.

**Funding:** We would like to thank the Forestry Public Welfare Scientific Research Project of China (Grant No. 201504303), the Central Public-interest Scientific Institution Basal Research Fund (Grant No. CAFYBB2019QD003) and the National Natural Science Foundations of China (Nos. 31570628 and 31570627) for the financial support of this study.

**Acknowledgments:** We are grateful to four anonymous reviewers, who provided the constructive comments and suggestions for improvement of the article.

**Data Accessibility:** Data used in this study are available from the Chinese Forestry Science Data Center (http://www.cfsdc.org/, accessed on 12 April 2020).

**Conflicts of Interest:** The authors declare no competing financial interests.

## Appendix A

**Table A1.** Performance of neural network with three hidden layers.

| Neurons of Each Layer | MSE | Iterations | MSE | Iterations | MSE | Iterations | MSE | Iterations |
|---|---|---|---|---|---|---|---|---|
| | tan:tan:tan | | tan:tan:log | | tan:log:tan | | tan:log:log | |
| 1:2:2:2:1 | 0.0884 | 26.2 | 0.0556 | 31.0 | 0.0940 | 33.6 | 0.1014 | 23.2 |
| 1:2:2:5:1 | 0.1020 | 18.2 | 0.0772 | 26.8 | 0.0505 | 19.4 | 0.0572 | 21.0 |
| 1:2:2:8:1 | 0.0461 | 24.0 | 0.0795 | 14.6 | 0.0878 | 17.8 | 0.0710 | 15.8 |
| 1:2:2:11:1 | 0.3880 | 23.8 | 0.0563 | 17.8 | 0.0945 | 21.6 | 0.0708 | 18.8 |
| 1:2:5:2:1 | 0.0674 | 45.0 | 0.0863 | 22.2 | 0.0800 | 17.2 | 0.1039 | 16.6 |
| 1:2:5:5:1 | 0.1028 | 14.6 | 0.1228 | 13.6 | 0.1759 | 25.4 | 0.0801 | 16.2 |
| 1:2:5:8:1 | 0.1194 | 24.2 | 0.0803 | 18.6 | 0.0518 | 15.4 | 0.1179 | 17.6 |
| 1:2:5:11:1 | 0.0761 | 18.4 | 0.0918 | 37.4 | 0.1213 | 20.4 | 0.0785 | 13.4 |
| 1:2:8:2:1 | 0.0776 | 24.8 | 0.1195 | 18.6 | 0.0511 | 21.8 | 0.0585 | 31.6 |
| 1:2:8:5:1 | 0.1066 | 16.4 | 0.0936 | 16.8 | 0.1049 | 17.2 | 0.1071 | 21.8 |
| 1:2:8:8:1 | 0.0560 | 19.2 | 0.0986 | 13.6 | 0.0545 | 15.8 | 0.0548 | 14.8 |
| 1:2:8:11:1 | 0.0788 | 43.2 | 0.0673 | 19.0 | 0.0899 | 15.8 | 0.0965 | 16.4 |
| 1:2:11:2:1 | 0.1351 | 19.4 | 0.1037 | 15.2 | 0.0803 | 25.6 | 0.0843 | 19.2 |
| 1:2:11:5:1 | 0.0772 | 15.0 | 0.0578 | 16.2 | 0.0621 | 19.0 | 0.0711 | 20.4 |
| 1:2:11:8:1 | 0.1403 | 40.2 | 0.1037 | 14.8 | 0.1418 | 16.0 | 0.1167 | 22.4 |
| 1:2:11:11:1 | 0.1180 | 25.4 | 0.0833 | 22.2 | 0.1200 | 25.4 | 0.1007 | 15.4 |
| 1:5:2:2:1 | 0.0660 | 18.4 | 0.1323 | 20.0 | 0.0446 | 23.2 | 0.0711 | 15.0 |
| 1:5:2:5:1 | 0.0624 | 16.4 | 0.0763 | 26.8 | 0.1192 | 25.0 | 0.0679 | 18.2 |
| 1:5:2:8:1 | 0.1065 | 27.8 | 0.1056 | 21.2 | 0.3581 | 17.8 | 0.0730 | 13.6 |
| 1:5:2:11:1 | 0.0892 | 19.2 | 0.0813 | 19.6 | 0.0796 | 22.0 | 0.1183 | 15.2 |
| 1:5:5:2:1 | 0.0630 | 20.2 | 0.0989 | 22.4 | 0.0733 | 16.4 | 0.1106 | 18.6 |
| 1:5:5:5:1 | 0.0625 | 14.6 | 0.0864 | 22.0 | 0.0981 | 17.0 | 0.0678 | 14.0 |
| 1:5:5:8:1 | 0.1087 | 19.4 | 0.1020 | 13.6 | 0.0501 | 13.8 | 0.0528 | 14.6 |
| 1:5:5:11:1 | 0.0537 | 20.2 | 0.0595 | 15.2 | 0.0634 | 14.2 | 0.1059 | 19.0 |
| 1:5:8:2:1 | 0.0716 | 13.6 | 0.0802 | 18.2 | 0.0752 | 20.6 | 0.0762 | 16.6 |
| 1:5:8:5:1 | 0.1409 | 13.4 | 0.1006 | 16.8 | 0.0676 | 18.2 | 0.0626 | 13.4 |
| 1:5:8:8:1 | 0.1101 | 18.8 | 0.1258 | 13.2 | 0.0652 | 15.6 | 0.0980 | 14.4 |
| 1:5:8:11:1 | 0.0590 | 16.8 | 0.1035 | 13.0 | 0.1775 | 16.6 | 0.1028 | 14.4 |
| 1:5:11:2:1 | 0.0665 | 17.0 | 0.0815 | 17.8 | 0.0509 | 16.0 | 0.0975 | 22.4 |
| 1:5:11:5:1 | 0.1387 | 14.2 | 0.0685 | 20.4 | 0.0796 | 14.4 | 0.1302 | 15.4 |
| 1:5:11:8:1 | 0.1145 | 12.8 | 0.0873 | 15.6 | 0.0919 | 17.8 | 0.0965 | 12.4 |
| 1:5:11:11:1 | 0.1451 | 15.2 | 0.0756 | 22.8 | 0.0856 | 12.8 | 0.0726 | 20.0 |
| 1:8:2:2:1 | 0.0709 | 15.8 | 0.0780 | 16.4 | 0.0689 | 15.0 | 0.0668 | 28.2 |
| 1:8:2:5:1 | 0.0814 | 34.8 | 0.0831 | 19.6 | 0.0698 | 16.4 | 0.0687 | 15.0 |
| 1:8:2:8:1 | 0.1178 | 19.8 | 0.0752 | 20.6 | 0.0593 | 16.8 | 0.0603 | 26.0 |
| 1:8:2:11:1 | 0.0814 | 17.6 | 0.0837 | 16.8 | 0.0580 | 19.4 | 0.0798 | 16.0 |
| 1:8:5:2:1 | 0.0811 | 15.6 | 0.1080 | 34.0 | 0.1530 | 18.4 | 0.0958 | 17.8 |
| 1:8:5:5:1 | 0.0916 | 15.4 | 0.0892 | 18.0 | 0.0653 | 13.6 | 0.1192 | 13.8 |
| 1:8:5:8:1 | 0.1110 | 13.8 | 0.0951 | 14.8 | 0.0814 | 13.6 | 0.1123 | 13.0 |
| 1:8:5:11:1 | 0.0899 | 15.8 | 0.0793 | 14.0 | 0.1076 | 12.6 | 0.0679 | 15.0 |
| 1:8:8:2:1 | 0.0939 | 15.6 | 0.1011 | 18.4 | 0.0859 | 14.8 | 0.1391 | 22.8 |
| 1:8:8:5:1 | 0.0772 | 18.0 | 0.1049 | 16.4 | 0.1353 | 14.6 | 0.1907 | 15.8 |
| 1:8:8:8:1 | 0.1859 | 14.0 | 0.0428 | 16.6 | 0.0793 | 13.2 | 0.1157 | 21.0 |
| 1:8:8:11:1 | 0.0812 | 16.2 | 0.1838 | 19.2 | 0.1607 | 14.2 | 0.0922 | 19.0 |
| 1:8:11:2:1 | 0.0957 | 22.8 | 0.0654 | 13.0 | 0.0578 | 21.6 | 0.1096 | 16.4 |
| 1:8:11:5:1 | 0.0544 | 13.0 | 0.0711 | 18.2 | 0.0579 | 16.2 | 0.0672 | 13.8 |
| 1:8:11:8:1 | 0.0937 | 12.4 | 0.1274 | 20.2 | 0.1468 | 17.0 | 0.0652 | 13.6 |
| 1:8:11:11:1 | 0.1193 | 23.2 | 0.1547 | 16.2 | 0.0652 | 15.4 | 0.0747 | 12.6 |
| 1:11:2:2:1 | 0.0767 | 16.6 | 0.0734 | 20.4 | 0.0953 | 17.2 | 0.0704 | 17.2 |
| 1:11:2:5:1 | 0.1564 | 13.8 | 0.0763 | 16.4 | 1.6445 | 21.2 | 0.0640 | 18.2 |
| 1:11:2:8:1 | 0.1009 | 17.4 | 0.0469 | 15.2 | 0.1369 | 12.4 | 0.1474 | 22.0 |

**Table A1.** *Cont.*

| Neurons of Each Layer | MSE | Iterations | MSE | Iterations | MSE | Iterations | MSE | Iterations |
|---|---|---|---|---|---|---|---|---|
| | tan:tan:tan | | tan:tan:log | | tan:log:tan | | tan:log:log | |
| 1:11:2:11:1 | 0.2392 | 42.4 | 0.1206 | 18.8 | 0.0919 | 18.6 | 0.0650 | 24.2 |
| 1:11:5:2:1 | 0.0780 | 26.4 | 0.0896 | 16.6 | 0.0711 | 16.6 | 0.1083 | 14.2 |
| 1:11:5:5:1 | 0.1551 | 13.8 | 0.1454 | 14.8 | 0.0846 | 13.8 | 0.0958 | 17.6 |
| 1:11:5:8:1 | 0.0814 | 15.6 | 0.0607 | 18.0 | 0.1620 | 15.8 | 0.0813 | 16.6 |
| 1:11:5:11:1 | 0.0987 | 15.0 | 0.0752 | 18.2 | 0.1069 | 13.4 | 0.1012 | 12.6 |
| 1:11:8:2:1 | 0.0901 | 16.0 | 0.0598 | 15.6 | 0.1069 | 14.6 | 0.0917 | 23.4 |
| 1:11:8:5:1 | 0.0838 | 17.0 | 0.1097 | 12.4 | 0.0570 | 15.8 | 0.0956 | 15.2 |
| 1:11:8:8:1 | 0.1041 | 14.4 | 0.1094 | 15.4 | 0.1474 | 15.2 | 0.1147 | 16.2 |
| 1:11:8:11:1 | 0.1105 | 21.8 | 0.1220 | 18.6 | 0.1508 | 15.0 | 0.0504 | 14.2 |
| 1:11:11:2:1 | 0.0755 | 18.4 | 0.0560 | 24.2 | 0.1350 | 12.0 | 0.0854 | 15.0 |
| 1:11:11:5:1 | 0.0785 | 18.6 | 0.0872 | 15.6 | 0.0854 | 11.8 | 0.1641 | 18.8 |
| 1:11:11:8:1 | 0.1129 | 16.0 | 0.0883 | 22.0 | 0.0946 | 13.2 | 0.1444 | 12.4 |
| 1:11:11:11:1 | 0.1004 | 14.6 | 0.1252 | 13.6 | 0.1538 | 13.2 | 0.0893 | 21.0 |
| 1:2:2:2:1 | 0.0985 | 20.6 | 0.0852 | 23.4 | 0.0840 | 42.4 | 0.1218 | 17.6 |
| 1:2:2:5:1 | 0.0861 | 66.2 | 0.1063 | 22.2 | 0.1230 | 18.4 | 0.0853 | 18.0 |
| 1:2:2:8:1 | 0.1157 | 19.2 | 0.0684 | 20.0 | 0.0812 | 40.2 | 0.1089 | 15.0 |
| 1:2:2:11:1 | 0.3540 | 26.6 | 0.0730 | 27.4 | 0.0613 | 22.0 | 0.0941 | 15.4 |
| 1:2:5:2:1 | 0.1066 | 29.6 | 0.0931 | 17.8 | 0.1062 | 17.4 | 0.3278 | 27.6 |
| 1:2:5:5:1 | 0.1151 | 17.2 | 0.0905 | 17.4 | 0.2043 | 16.8 | 0.0690 | 14.4 |
| 1:2:5:8:1 | 0.0749 | 18.2 | 0.0859 | 23.8 | 0.0686 | 17.0 | 0.0782 | 14.4 |
| 1:2:5:11:1 | 0.0578 | 15.0 | 0.0469 | 17.4 | 0.1966 | 18.0 | 0.1248 | 21.0 |
| 1:2:8:2:1 | 0.1176 | 21.6 | 0.0458 | 19.2 | 0.0827 | 19.2 | 0.0691 | 21.0 |
| 1:2:8:5:1 | 0.0885 | 17.6 | 0.0719 | 17.2 | 0.1539 | 16.8 | 0.0821 | 20.6 |
| 1:2:8:8:1 | 0.0824 | 25.0 | 0.1182 | 14.6 | 0.0804 | 21.4 | 0.0681 | 18.0 |
| 1:2:8:11:1 | 0.0797 | 19.6 | 0.0807 | 17.0 | 0.0580 | 13.2 | 0.0983 | 17.2 |
| 1:2:11:2:1 | 0.1024 | 15.6 | 0.1074 | 18.2 | 0.0727 | 32.4 | 0.0829 | 17.2 |
| 1:2:11:5:1 | 0.0961 | 15.4 | 0.0696 | 16.6 | 0.0785 | 15.0 | 0.0819 | 23.0 |
| 1:2:11:8:1 | 0.0922 | 19.6 | 0.0723 | 19.0 | 0.0921 | 15.6 | 0.0797 | 30.0 |
| 1:2:11:11:1 | 0.0733 | 19.8 | 0.0872 | 13.8 | 0.0734 | 15.2 | 0.1016 | 14.4 |
| 1:5:2:2:1 | 0.0874 | 13.4 | 0.0864 | 23.8 | 0.0673 | 24.0 | 0.0782 | 16.6 |
| 1:5:2:5:1 | 0.0769 | 17.8 | 0.0715 | 19.2 | 0.0668 | 15.6 | 0.1170 | 15.8 |
| 1:5:2:8:1 | 0.0651 | 14.0 | 0.0718 | 20.8 | 0.0729 | 19.8 | 0.1238 | 18.8 |
| 1:5:2:11:1 | 0.0661 | 19.0 | 0.0894 | 14.6 | 0.0793 | 14.4 | 0.1327 | 15.0 |
| 1:5:5:2:1 | 0.1260 | 15.4 | 0.1682 | 15.6 | 0.0627 | 17.8 | 0.1830 | 14.8 |
| 1:5:5:5:1 | 0.1038 | 13.4 | 0.0750 | 15.8 | 0.0748 | 18.8 | 0.0991 | 16.8 |
| 1:5:5:8:1 | 0.0684 | 13.0 | 0.0522 | 33.8 | 0.0786 | 17.6 | 0.0999 | 15.2 |
| 1:5:5:11:1 | 0.1568 | 17.4 | 0.0920 | 18.0 | 0.0900 | 18.8 | 0.1166 | 19.2 |
| 1:5:8:2:1 | 0.0793 | 23.4 | 0.0947 | 17.4 | 0.0521 | 14.0 | 0.1257 | 19.4 |
| 1:5:8:5:1 | 0.1234 | 21.6 | 0.1050 | 19.0 | 0.0716 | 19.2 | 0.1332 | 21.6 |
| 1:5:8:8:1 | 0.0903 | 14.0 | 0.0794 | 11.6 | 0.1379 | 19.0 | 0.0695 | 17.4 |
| 1:5:8:11:1 | 0.1751 | 17.4 | 0.1073 | 18.2 | 0.1630 | 21.2 | 0.0895 | 13.2 |
| 1:5:11:2:1 | 0.0927 | 18.2 | 0.1065 | 15.4 | 0.0739 | 32.4 | 0.1177 | 24.8 |
| 1:5:11:5:1 | 0.0941 | 13.2 | 0.0847 | 16.2 | 0.0821 | 21.2 | 0.0702 | 13.8 |
| 1:5:11:8:1 | 0.2084 | 20.6 | 0.0703 | 13.6 | 0.0994 | 14.0 | 0.1120 | 22.0 |
| 1:5:11:11:1 | 0.0896 | 22.0 | 0.0832 | 21.2 | 0.0895 | 14.2 | 0.0853 | 13.2 |
| 1:8:2:2:1 | 0.1322 | 14.8 | 0.1244 | 16.0 | 0.0948 | 16.0 | 0.0940 | 17.6 |
| 1:8:2:5:1 | 0.0963 | 13.0 | 0.0981 | 16.0 | 0.1278 | 21.8 | 0.0717 | 16.8 |
| 1:8:2:8:1 | 0.1204 | 15.0 | 0.0961 | 15.8 | 0.0900 | 29.2 | 0.0823 | 19.8 |
| 1:8:2:11:1 | 0.0550 | 22.4 | 0.1335 | 12.8 | 0.1197 | 13.2 | 0.0694 | 14.2 |
| 1:8:5:2:1 | 0.0824 | 17.2 | 0.0804 | 23.6 | 0.1899 | 22.6 | 0.0671 | 18.4 |
| 1:8:5:5:1 | 0.0693 | 17.6 | 0.1303 | 20.4 | 0.0820 | 15.6 | 0.0867 | 15.4 |
| 1:8:5:8:1 | 0.0930 | 14.8 | 0.1011 | 13.2 | 0.1109 | 12.6 | 0.1415 | 15.0 |
| 1:8:5:11:1 | 0.2495 | 17.4 | 0.0850 | 13.2 | 0.0676 | 15.2 | 0.1607 | 13.0 |

**Table A1.** *Cont.*

| Neurons of Each Layer | MSE | Iterations | MSE | Iterations | MSE | Iterations | MSE | Iterations |
|---|---|---|---|---|---|---|---|---|
| | tan:tan:tan | | tan:tan:log | | tan:log:tan | | tan:log:log | |
| 1:8:8:2:1 | 0.0538 | 15.2 | 0.0825 | 21.6 | 0.1115 | 18.0 | 0.1038 | 25.6 |
| 1:8:8:5:1 | 0.0827 | 12.4 | 0.0513 | 15.4 | 0.0678 | 14.2 | 0.1517 | 16.4 |
| 1:8:8:8:1 | 0.1120 | 17.0 | 0.0994 | 15.6 | 0.1560 | 15.0 | 0.0733 | 12.2 |
| 1:8:8:11:1 | 0.1027 | 17.4 | 0.1331 | 13.4 | 0.1040 | 14.6 | 0.0763 | 21.0 |
| 1:8:11:2:1 | 0.0499 | 15.0 | 0.0648 | 18.6 | 0.0955 | 20.0 | 0.0762 | 29.6 |
| 1:8:11:5:1 | 0.1685 | 16.4 | 0.1082 | 13.0 | 0.0982 | 15.6 | 0.0786 | 12.4 |
| 1:8:11:8:1 | 0.0700 | 13.2 | 0.1409 | 12.4 | 0.1046 | 19.0 | 0.1053 | 22.6 |
| 1:8:11:11:1 | 0.1234 | 18.2 | 0.0767 | 18.2 | 0.0686 | 16.6 | 0.0789 | 19.2 |
| 1:11:2:2:1 | 0.0864 | 19.2 | 0.0585 | 20.2 | 0.0655 | 20.4 | 0.0920 | 15.0 |
| 1:11:2:5:1 | 0.2456 | 17.2 | 0.1119 | 16.4 | 0.0881 | 23.6 | 0.1038 | 14.8 |
| 1:11:2:8:1 | 0.0570 | 19.2 | 0.0852 | 21.8 | 0.0724 | 17.8 | 0.1493 | 26.4 |
| 1:11:2:11:1 | 0.1345 | 18.2 | 0.1407 | 19.2 | 0.1392 | 14.4 | 0.0836 | 16.2 |
| 1:11:5:2:1 | 0.1205 | 18.6 | 0.0658 | 20.6 | 0.0561 | 16.8 | 0.1163 | 16.8 |
| 1:11:5:5:1 | 0.0736 | 17.2 | 0.0896 | 17.2 | 0.0867 | 17.0 | 0.1203 | 23.6 |
| 1:11:5:8:1 | 0.0746 | 15.6 | 0.1327 | 16.4 | 0.0625 | 20.0 | 0.1371 | 17.2 |
| 1:11:5:11:1 | 0.0634 | 14.6 | 0.0999 | 18.4 | 0.0862 | 15.6 | 0.1830 | 19.8 |
| 1:11:8:2:1 | 0.0918 | 19.4 | 0.1124 | 25.2 | 0.1609 | 13.8 | 0.0704 | 16.4 |
| 1:11:8:5:1 | 0.2184 | 19.4 | 0.0709 | 14.0 | 0.1025 | 14.6 | 0.1356 | 15.6 |
| 1:11:8:8:1 | 0.1078 | 17.8 | 0.0873 | 14.4 | 0.0806 | 15.4 | 0.1091 | 13.0 |
| 1:11:8:11:1 | 0.0693 | 15.4 | 0.1357 | 16.6 | 0.0812 | 14.2 | 0.1164 | 16.0 |
| 1:11:11:2:1 | 0.0992 | 15.2 | 0.0919 | 14.8 | 0.1202 | 16.4 | 0.0973 | 14.0 |
| 1:11:11:5:1 | 0.0992 | 16.8 | 0.1095 | 13.6 | 0.0990 | 17.2 | 0.0652 | 13.6 |
| 1:11:11:8:1 | 0.1335 | 19.4 | 0.0798 | 17.2 | 0.0928 | 15.8 | 0.0991 | 17.0 |
| 1:11:11:11:1 | 0.1030 | 13.8 | 0.1092 | 22.0 | 0.1025 | 14.8 | 0.1140 | 12.6 |

**Table A2.** Parameter estimates of traditional height-diameter models.

| Equations | Parameter Estimates | | |
|---|---|---|---|
| | a | b | c |
| Richards | 13.2393 | 0.1119 | 1.9216 |
| Logistic | 12.3167 | 7.0594 | 0.1959 |
| Gompertz | 12.6456 | 3.0414 | 0.1447 |
| Korf | 16.9864 | 11.9539 | 1.0749 |
| Mitscherlich | 22.2102 | 0.0331 | |
| Schumacher | 18.0091 | 10.6440 | |

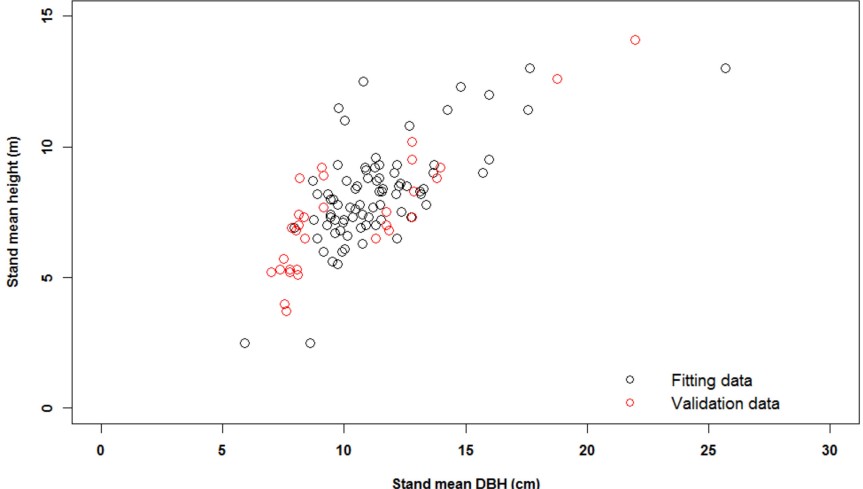

**Figure A1.** Scattered diagram of stand mean height and stand mean diameter at breast height (DBH) for both model fitting and validation datasets.

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
