# Peer review of "Modeling Height–Diameter Relationship for Poplar Plantations Using Combined-Optimization Multiple Hidden Layer Back Propagation Neural Network"

_forests, doi:10.3390/f11040442_

Round 1
Reviewer 1 Report
Title: Modelling height-diameter relationship for poplar plantation using combined-optimization multiple hidden layer back propagation neural networks
Authors: Jianbo Shen et al.
This study reports a neural network approach for modelling tree height-diameter relationships using poplar plantations in China. The study shows the advantage of multiple hidden layered neural network models comparing with single hidden layered neural network models and classical nonlinear least-squares regressions. The purposes of the study are clear and meet reader’s interests. However, there are some points to be revised, especially in terms of the comparison between neural network models and classical regressions.
General points
- Mixed effects models should be included in the performance comparisons
In this study, nonlinear least-squares regressions are used for performance comparisons with neural network models. I wonder why the authors did not include mixed-effects models in the comparisons. Özçelik et al. (2013) concluded that when the variability of the height-diameter relationship from stand to stand can be incorporated into the model, both mixed-effects nonlinear regressions and neural networks approaches are useful tools for tree height prediction. As the authors mentioned in Discussion (L396-399), mixed-effects models have a potential to provide higher accuracy that may be comparable to neural network models. Currently, mixed-effects models are very popular and used by many researchers and students. It’s worthy of including mixed-effects models in the comparisons to clearly show the advantage of neural network models.
Özçelik et al. (2013) For Ecol Manage 306: 52-60.
- Shortcomings and disadvantages should also be described for objective evaluation
In this manuscript, short comings and disadvantages of neural network models are not clearly pointed out. In view of fair and objective comparisons of the models, such shortcomings should be described. In the same way, advantages of traditional models and mixed-effects models (for example, easy to use and easy to understand the model formula and the effects of parameters) should also be provided.
Specific points
L4 and many others, “Propagrtation” -> “Propagation” many typos are found in the manuscript. Check all the typos (using spell checking tools) and correct them.
L22-28, Detailed descriptions of the methods are not necessary in Abstract and should be removed.
L47-49, “It is difficult to measure tree height directly …” This sentence looks redundant and is not necessary.
L84-93 “Javier Castano-Santamaria et al. [14] predicted ... “ Sentences of previous studies look redundant. The explanation is not necessarily provided for each of these studies. Instead, the studies would be better to be collectively described from several points of view such as technical aspects and forest types.
L122-123 “sample plots surveyed in 1997 and 2002 … consisted of a total of 96 sample plots” Does this mean “consisted of 96 sample plots for each year”? Was the survey in 2002 conducted in the same sample plots as in 1997? If so, they cannot be simply combined because the data taken in the same plot in different years are not independent each other.
L124-125, “A total of 112 data records were …” Is this correct? Isn’t it 116 data records in total (20 + 96 = 116)? As mentioned above, combining the data in 1997 and 2002 might not be appropriate at least for traditional least squares regressions because they are not independent.
L173, “logsig and tansig”, First, spell out all terms as “logistic sigmoid” and “tangent sigmoid”.
L173, “The former is a unipolar S-function …” Put the equation numbers (Equations 3 and 4).
L265, “with use of k = 5” How was the value (k=5) determined?
Tables 3 and 4 What do the horizontal lines on the first row of values represent?
L292 and Fig. 2 What is bias? Explanations of bias in neural network should be needed.
L307, “24.4%” -> “21.0%? Check the calculation.
L338, 362 “performance-price”, “performance-price ratio” Meaning is unclear. Can these terms be rephrased as just “performance”?
L348 “however” This word is not necessary here and should be removed.
L385, “We applied the same method, …” Meaning is unclear. What is the same method?
L388, “provides a reference basis for this” Meaning is unclear. The terms should be rephrased.
L391-392, “Only when the same inputs are considered … more meaningful” This sentence is not necessary because the earlier sentence (L388, “The comparison makes the sense when …”) is almost the same.
L396-399, “Some other modelling approaches, …” As the authors mentioned here, neural network should be compared with mixed-effects models.
L399-401, “all potential factors .. adequately described” How are the factors adequately described in the neural network modelling? Can the authors provide more details?
L403-407 “This study proposes …” This paragraph is not necessary and should be removed because similar sentences are provided in Conclusions.
L417, “about 6% higher” -> “8.5% higher”?
L419-421, “tree height growth is also significantly affected by … such as site, climate and competition factors …” Citation of some literature related to the effects of these factors is preferable if possible.
Author Response
This study reports a neural network approach for modelling tree height-diameter relationships using poplar plantations in China. The study shows the advantage of multiple hidden layered neural network models comparing with single hidden layered neural network models and classical nonlinear least-squares regressions. The purposes of the study are clear and meet reader’s interests. However, there are some points to be revised, especially in terms of the comparison between neural network models and classical regressions.
Response: Thanks for your positive comments on our manuscript (MS). The comments and suggestions that you gave are all valuable and helpful for revising and improving our MS. We have read the comments and suggestions carefully and made the corresponding corrections and clarifications.
General points
- Mixed effects models should be included in the performance comparisons
In this study, nonlinear least-squares regressions are used for performance comparisons with neural network models. I wonder why the authors did not include mixed-effects models in the comparisons. Özçelik et al. (2013) concluded that when the variability of the height-diameter relationship from stand to stand can be incorporated into the model, both mixed-effects nonlinear regressions and neural networks approaches are useful tools for tree height prediction. As the authors mentioned in Discussion (L396-399), mixed-effects models have a potential to provide higher accuracy that may be comparable to neural network models. Currently, mixed-effects models are very popular and used by many researchers and students. It’s worthy of including mixed-effects models in the comparisons to clearly show the advantage of neural network models.
Özçelik et al. (2013) For Ecol Manage 306: 52-60.
Response:Thank you very much for this suggestion. Our main intention was that this article should provide the insights about the methods of choosing the best neural networks structure for an optimal H-D model and slight comparison of this model against H-D model resulted from traditional modeling approach. However, considering your suggestion, we also developed the mixed-effects model and compared its results against those from the artificial neural networks approach. We have added a new sub-section “3.4 Comparison to the mixed effects model accuracy” in the revised manuscript. We also briefly described the mixed-effect modeling in sub-section 2.2.2 as well. We found that fit statistics of the mixed effects H-D model slightly inferior to the BP neural network H-D model, but substantially superior to the height-diameter model fitted using ordinary least square regression approach. We added a brief description of the comparison of mixed-effects model and BP neural networks height-diameter model in 2.2.2 sub-section.
- Shortcomings and disadvantages should also be described for objective evaluation
In this manuscript, short comings and disadvantages of neural network models are not clearly pointed out. In view of fair and objective comparisons of the models, such shortcomings should be described. In the same way, advantages of traditional models and mixed-effects models (for example, easy to use and easy to understand the model formula and the effects of parameters) should also be provided.
Response:Thanks for suggestion. Generally speaking, neural network models have strong robustness, and the parameters of traditional models and mixed effects models have biological significance. The neural network weights and thresholds are not easy to explain, but the fitting accuracy of the neural network is higher than both the traditional models and mixed effects model. These information was added in better way in the discussion section of the revised MS.
Specific points
- L4 and many others, “Propagrtation” -> “Propagation” many typos are found in the manuscript. Check all the typos (using spell checking tools) and correct them.
Response: We checked carefully and corrected throughout the MS.
- L22-28, Detailed descriptions of the methods are not necessary in Abstract and should be removed.
Response:Thanks for this suggestion. We shorted this in the revised MS as much as we could. Whatever we have now, it is considered necessary to understand about the study clearly.
- L47-49, “It is difficult to measure tree height directly …” This sentence looks redundant and is not necessary.
Response:Thank for this notice. We deleted this sentence and slightly rephrased the texts.
- L84-93 “Javier Castano-Santamaria et al. [14] predicted ... “ Sentences of previous studies look redundant. The explanation is not necessarily provided for each of these studies. Instead, the studies would be better to be collectively described from several points of view such as technical aspects and forest types.
Response:Thanks for your suggestion. We agreed on your idea and listed individually who did what very briefly and briefly summarized them based on features of their methods in overall. Thus, it seems that text here is concise but clear.
- L122-123 “sample plots surveyed in 1997 and 2002 … consisted of a total of 96 sample plots” Does this mean “consisted of 96 sample plots for each year”? Was the survey in 2002 conducted in the same sample plots as in 1997? If so, they cannot be simply combined because the data taken in the same plot in different years are not independent each other.
Response:Thanks for this concern. A total of 9659 trees in 112 sample plots (of which 20 sample plots were measured in 1997 and 92 sample plots in 2002) were utilized for modeling height-diameter relationship”, which is in the material sub-section, please see there. All sample plots were mutually independent and not repeatedly measured, which did not cause dependency among observations. In addition, all sample plots designed in this study were very representative to the entire poplar (Populus spp. L.) plantations in the Guangdong province of China. Although the current data set used in this study is not big, resulting model is enough representative for other similar forests where the same species is present. This article emphasizes more on the methodology, aiming that it may be useful for other researchers to develop similar BP neural network height-diameter models for other species as well.
- L124-125, “A total of 112 data records were …” Is this correct? Isn’t it 116 data records in total (20 + 96 = 116)? As mentioned above, combining the data in 1997 and 2002 might not be appropriate at least for traditional least squares regressions because they are not independent.
Response:Thanks. We made mistake here. There are 112 sample plots which contain 20 sample plots data records in 1997 and 92 sample plots data records in 2002. We modified the data materials sub-section substantially. Now everything is clearly understandable.
- L173, “logsig and tansig”, First, spell out all terms as “logistic sigmoid” and “tangent sigmoid”.
Response:Thank you. It was corrected in the revised MS.
- L173, “The former is a unipolar S-function …” Put the equation numbers (Equations 3 and 4).
Response:Thanks. It was corrected in the revised MS.
- L265, “with use of k = 5” How was the value (k=5) determined?
Response:This was chosen from our experience of k-fold cross validation, in which and 5 and 10 are commonly used in existing studies. It doesn't exceed 10, considering that 10 takes a longer time for convergence of the neural networks and 5 could take shorter time. We also follow the recommendation by Arlot and Lerasle [40] recommended. It was clarified in the revised MS.
- Tables 3 and 4 What do the horizontal lines on the first row of values represent?
Response:It was revised in the revised MS.
- L292 and Fig. 2 What is bias? Explanations of bias in neural network should be needed.
Response: Thanks. The purpose of offset bias in the network is to make offset compensation when the input distribution of a layer is not 0 as the center. If we miss the bias term when we write neural network codes, the neural network is likely to become very poor, with slow convergence and poor accuracy, and may even fail to converge.
- L307, “24.4%” -> “21.0%? Check the calculation.
Response:Thank you. According to suggestion from reviewer # 1, each traditional H-D function was added with 1.3, re-estimated, and found significant changes in the results and all corrections were applied wherever necessary.
- L338, 362 “performance-price”, “performance-price ratio” Meaning is unclear. Can these terms be rephrased as just “performance”?
Response:Done.
- L348 “however” This word is not necessary here and should be removed.
Response:Done.
- L385, “We applied the same method, …” Meaning is unclear. What is the same method?
Response:Rephrased.
- L388, “provides a reference basis for this” Meaning is unclear. The terms should be rephrased.
Response:Rephrased.
- L391-392, “Only when the same inputs are considered … more meaningful” This sentence is not necessary because the earlier sentence (L388, “The comparison makes the sense when …”) is almost the same.
Response:Deleted.
- L396-399, “Some other modelling approaches, …” As the authors mentioned here, neural network should be compared with mixed-effects models.
Response:We did this according to your suggestions. Here we modified the text accordingly.
- L399-401, “all potential factors. adequately described” How are the factors adequately described in the neural network modelling? Can the authors provide more details?
Response:Thank you. We assumed that influences of various factors on the height –diameter relationships are similar as those of the applying different transfer functions, number of hidden layers and neurons of each hidden layer on the accuracy of neural network height-diameter models. The stated text is our assumptions, but need serious investigations on such. We slightly modified the text here, but the sense remains almost the same.
- L403-407 “This study proposes …” This paragraph is not necessary and should be removed because similar sentences are provided in Conclusions.
Response:We deleted.
- L417, “about 6% higher” -> “8.5% higher”?
Response:This was changed according to new results obtained from refitting of the models according to suggestion from reviewer # 1.
- L419-421, “tree height growth is also significantly affected by … such as site, climate and competition factors …” Citation of some literature related to the effects of these factors is preferable if possible.
Response:Thanks for this suggestion. However, we considered not mentioning any citation here in the conclusion section, because this section needs to be made easily readable and understandable independently without the support of any other sections of this article and any references.
Reviewer 2 Report
Topic of the manuscript is interesting and its novelty is in using of multiple hidden layer back propagation neural networks. Manuscript needs more care about English and mainly it need to make a better discussion – it is the weakest part of the manuscript. Another problem is using of traditional forest growth models. If you want to fit relationship between tree height and diameter at breast height, you have to use height-diameter (H-D) models. But H-D models are not the growth models – they are similar, but not the same. Do not use term “growth models” in the whole manuscript.
Title
Page 1, line 4: Replace “Propagrtation“ with ”Propagation“.
Abstract:
Page 1, line 16: Replace “diammeter” with “diameter”.
Page 1, line 23: Add “at” to diameter breast height.
Page 1, line 28: Replace “(PopulusL) with “(Populus L.)”.
Page 1, line 31: Replace “structre” with “structure”.
Page 1, line 37: Replace “neuran” with “neural”.
Keywords:
Replace “traditional growth functions” with traditional height-diameter functions”.
Add “s” to Richard functions.
Introduction
Page 2, line 48: Replace “realively” with “relatively”.
Page 2, line 59: Replace “site qulity” with “site quality”.
Page2, line 65: Replace “contituents” with “constituents”.
Page 2, line 67: Replace “desnsity” with “density”.
Page 2, line 70: Replace “addion” with “addition”.
Page 2, line 84: Delete “Javier”. It is only first name of the author.
Page 3, line 103: Add “of” after “robustness”.
Materials and Methods
Section 2.1. Data materials
From this section is not clearly evident how many trees or how many sample plots were measured. Here is only written that a total of 112 data records were obtained, but is not defined what data record is – individual tree or sample plot?
Please add the area and the shape of sample plot too.
Section 2.2.1. Traditional approach
As I wrote above, do not use growth functions but H-D functions. Your equations of the models must content constant 1.3 before parameter a. Reason is simple – growth function starts in the coordinates [0, 0] in the XY plot but the H-D function starts in the coordinates [0, 1.3].
Page 3, line 139: Replace “functuons” with “functions”.
Page 3, line 151: Replace “netwrks” with “networks”.
Page 3, line 166: Replace “alterantives” with “alternatives”.
Equation 3 and 4: They are not connected with the text and the explanation of their reason is missing.
Page 5, line 215: Add “of” after “coefficient”.
Results
Page 8, lines 278-279: Do not use term “significant” when did you not make statistical test. Or if you made the test, write the results of the test here. Apply this recommendation to whole manuscript.
Page 9, line 299: Here is only one place, where you used the right name of the model - height-diameter model.
Table 6: Because you have to refit your traditional model with constant 1.3 so you will have got new results of evaluation indices and model parameters (Table 2 in Appendix). You have to rewrite your results in the text and in the tables 6 and 2 (in appendix) too.
Page 9, line 318: Replace “perfronace” with “performance”.
Discussion
As I wrote above, the discussion is the weakest part of the manuscript and must be overhauled. It is full of your personal statement without comparison with similar research. I understand that your research is new and unique, but I think that lot of your general notes and statements can be connected with at least one references.
Lines 341-343: Here you wrote that only few studies have existed – so find them and work with them in the discussion.
Page 10, line 330: Replace “pedcition” with “prediction”.
Page 10, line 347: Table 8 does not exist.
Page 11, line 361: Replace “modelers finding” with “modellers to find”.
Page 11, line 368: Replace “patetrns” with “paterns”.
Conclusions
Page 12, line 411: Replace “predciton” with “prediction”.
Page 12, lines 420-421. The sentence “,which will be added in the subsequent modelling studies to improve the height prediction accuracies.” should be deleted.
References
Delete symbol “[J]” from all of references.
Here is lot of mistakes in the reference list. Number 5 and 13 are the same references, number 9 and 23 are the same references too. Another problem is in authors with two surnames – the break between these two surnames is missing – for example Crecentecampo F. is Crecente Campo F. and so on. Take more care to reference list and mainly to first names and surnames of authors.
Author Response
Topic of the manuscript is interesting and its novelty is in using of multiple hidden layer back propagation neural networks. Manuscript needs more care about English and mainly it need to make a better discussion – it is the weakest part of the manuscript. Another problem is using of traditional forest growth models. If you want to fit relationship between tree height and diameter at breast height, you have to use height-diameter (H-D) models. But H-D models are not the growth models – they are similar, but not the same. Do not use term “growth models” in the whole manuscript.
Response: Many thanks for your positive comments on our MS. We edited the English language very carefully throughout the manuscript, and also improved the discussion section considerably as per your and other reviewers’ suggestions. We agree on your view that H-D model is different from height growth models. We therefore used the terminology cautiously following your suggestions.
Title:
- Page 1, line 4: Replace “Propagrtation“ with ”Propagation“.
Response: Done.
Abstract:
- Page 1, line 16: Replace “diammeter” with “diameter”.
Response: Done.
- Page 1, line 23: Add “at” to diameter breast height.
Response: Done.
- Page 1, line 28: Replace “(PopulusL) with “(Populus L.)”.
Response: Done.
- Page 1, line 31: Replace “structre” with “structure”.
Response: Done.
- Page 1, line 37: Replace “neuran” with “neural”.
Response: Done.
Keywords:
- Replace “traditional growth functions” with traditional height-diameter functions”.
Add “s” to Richard functions.
Response: Done as suggested.
Introduction
- Page 2, line 48: Replace “realively” with “relatively”.
Response: Done.
- Page 2, line 59: Replace “site qulity” with “site quality”.
Response: Done.
- Page2, line 65: Replace “contituents” with “constituents”.
Response: Done.
- Page 2, line 67: Replace “desnsity” with “density”.
Response: Done.
- Page 2, line 70: Replace “addion” with “addition”.
Response: Done.
- Page 2, line 84: Delete “Javier”. It is only first name of the author.
Response: Done.
- Page 3, line 103: Add “of” after “robustness”.
Response: Done.
Materials and Methods
Section 2.1. Data materials
- From this section is not clearly evident how many trees or how many sample plots were measured. Here is only written that a total of 112 data records were obtained, but is not defined what data record is – individual tree or sample plot?
Response: Thanks for pointing out this problem here. We used data from 9659 trees on 112 sample plot as a final modeling data set. We substantially modified the texts here in order to make the message clearer.
- Please add the area and the shape of sample plot too.
Response: The shape of sample plots is square and area of which is 666.67 m2. This information was added in the revised MS.
Section 2.2.1. Traditional approach
- As I wrote above, do not use growth functions but H-D functions. Your equations of the models must content constant 1.3 before parameter a. Reason is simple – growth function starts in the coordinates [0, 0] in the XY plot but the H-D function starts in the coordinates [0, 1.3].
Response: Thanks for pointing out this problem here. We replaced the term “growth function” with “height-diameter function” where applicable and added 1.3 to the equations listed in Table 2, and refitted each of them to the data.
- Page 3, line 139: Replace “functuons” with “functions”.
Response: Done.
- Page 3, line 151: Replace “netwrks” with “networks”.
Response: Done.
- Page 3, line 166: Replace “alterantives” with “alternatives”.
Response: Done.
- Equation 3 and 4: They are not connected with the text and the explanation of their reason is missing.
Response: Thanks. We have defined them properly in the revised MS.
- Page 5, line 215: Add “of” after “coefficient”.
Response: Done.
Results
- Page 8, lines 278-279: Do not use term “significant” when did you not make statistical test. Or if you made the test, write the results of the test here. Apply this recommendation to whole manuscript.
Response: Thanks for this suggestion, and we applied this recommendation where applicable.
- Page 9, line 299: Here is only one place, where you used the right name of the model - height-diameter model.
Response: Thanks for your approval. We have applied this terminology throughout the manuscript.
- Table 6: Because you have to refit your traditional model with constant 1.3 so you will have got new results of evaluation indices and model parameters (Table 2 in Appendix). You have to rewrite your results in the text and in the tables 6 and 2 (in appendix) too.
Response: Thanks for this suggestion. We refitted all traditional height-diameter models and neural networks height-diameter models and found slight change of the results, and thus updated the results were presented in the revised MS.
- Page 9, line 318: Replace “perfronace” with “performance”.
Response: Done.
Discussion
- As I wrote above, the discussion is the weakest part of the manuscript and must be overhauled. It is full of your personal statement without comparison with similar research. I understand that your research is new and unique, but I think that lot of your general notes and statements can be connected with at least one references.
Response: Thanks for this important suggestion. We compared the results from our BP neural networks height-diameter model against those of more or less similar existing studies in our discussion section. For example, we attempted to compare our results against those from Özçelik et al. (17), Castaño-Santamaría et al. (13), and Castro et al. (18), which are substantially different from our BP neural networks modeling, as they all compared the precision with different input variables, did not compare different transfer functions and hidden layers and so on.
- Lines 341-343: Here you wrote that only few studies have existed – so find them and work with them in the discussion.
Response:Many thanks for this important concern here. We regret for our mistake that we made here. To the authors’ knowledge, none of the studies conducted exactly the same ways as ours so far. We rephrased the texts here accordingly and added one more sentence here in favor of this message.
- Page 10, line 330: Replace “pedcition” with “prediction”.
Response: Done.
- Page 10, line 347: Table 8 does not exist.
Response: Thanks. Indeed Table 8 is the Table 6, we corrected this.
- Page 11, line 361: Replace “modelers finding” with “modellers to find”.
Response: Done.
- Page 11, line 368: Replace “patetrns” with “paterns”.
Response: Done.
Conclusions
- Page 12, line 411: Replace “predciton” with “prediction”.
Response: Done.
- Page 12, lines 420-421. The sentence “,which will be added in the subsequent modelling studies to improve the height prediction accuracies.” should be deleted.
Response: Done.
References
- Delete symbol “[J]” from all of references.
Response:Done.
- Here is lot of mistakes in the reference list. Number 5 and 13 are the same references, number 9 and 23 are the same references too. Another problem is in authors with two surnames – the break between these two surnames is missing – for example Crecentecampo F. is Crecente Campo F. and so on. Take more care to reference list and mainly to first names and surnames of authors.
Response:Thanks for pointing out this problem. Repeated references were deleted in the revised MS. and we carefully checked all the references and if found any mistake, they were corrected.
Reviewer 3 Report
Review on
Modelling height-diameter relationship for popular plantations using combined-optimization multiple hidden layer back propagation neural networks
This paper compares fitting height-diameter relationships with neural networks to six traditional methods. Different ways to fit the neural network were systematically tested and described. The paper reads well and cites the relevant literature. My major criticism with the paper is the comparatively small data set of dbh and height measurements used, since this type of data is not hard to acquire. Despite of this I think it is a very interesting paper that shows the potential of machine learning for forestry applications. Another draw-back of this paper is the very large number of typos or small grammatical errors. My review term is minor revisions, but the paper should be send to a professional English proof reading service (I am not an English native speaker) before publication.
Detailed comments:
Line 31: This is not clear. The best structure determined was a model with two hidden-layers with 5 neurons in each layer
Line 32: List both transfer functions: logsig, tansig, - Maybe also the functions could be spelled out – log sig is probably the logistic transfer function
Line 79-82: This is too vague, please be more specific
Line 122: Sample plots or trees samples? If it is plots please give the number of trees.
Line 124-125: 20 and 96 does not sum up to 112 are these trees
Line 169: Please give the reference for the formula
Line 173, 174: The reference to equation 3 and 4 is missing, logsig – is this the logistic function, which is given in equation 4?
Line 189-190: strictly speaking mapminmax scales inputs so it always falls in the interval [-1,1]
Line 229: What was k in your study?
Line 230-249: It might not be necessary to describe the k-fold cross-validation algorithm. Also I think there is a problem with the indices in this section: Line 236: Model Mj; Line 242: What does the index p stand for?
Line 274-276: This is not clear to my
Table 5: There seem to be some lines which are duplicated
Line 315: In the first equation the slope coefficient seems to be missing
Line 316: I do not understand this sentence
Line 320: Why did you choose -1.1 m and -2.2 m?
Line 338: I don’t think performance-price is the correct expression here.
Line 339: Do you think your results are generalizable for different sample sizes?
Line 363: Do such interfaces exist in standard statistical software?
Line 382: What do you mean by unified method?
Typos
Line 4: propagation
Line 28: Populus spp. L.
Line 37: neural
Line 47: equations
Line 48: relatively
Line 52: the optimal
Line 54: the main theme
Line 59: site quality
Line 65: constituents
Line 83: Maybe better: giving precise results
…Please send out for English proof reading
Author Response
This paper compares fitting height-diameter relationships with neural networks to six traditional methods. Different ways to fit the neural network were systematically tested and described. The paper reads well and cites the relevant literature. My major criticism with the paper is the comparatively small data set of dbh and height measurements used, since this type of data is not hard to acquire. Despite of this I think it is a very interesting paper that shows the potential of machine learning for forestry applications. Another draw-back of this paper is the very large number of typos or small grammatical errors. My review term is minor revisions, but the paper should be send to a professional English proof reading service (I am not an English native speaker) before publication.
Response: Many thanks for evaluating our manuscript and providing the suggestions and recommendations to improve the MS quality that will have a good novelty. We revised the MS according to your suggestions and recommendations carefully, which resulted in a substantially improved MS. All sample plots designed in this study were very representative to the entire poplar (Populus spp. L.) plantations in the Guangdong province of China. Although the current data set used in this study is not big, resulting model is enough representative for other similar forests where the same species is present. In addition, this article emphasizes more on the methodology, aiming that it may be useful for other researchers to develop similar BP neural network height-diameter models for other species as well. We also edited the English language very carefully throughout the revised MS. The current version of the manuscript is ready to be published.
Detailed comments:
- Line 31: This is not clear. The best structure determined was a model with two hidden-layers with 5 neurons in each layer
Response: Thanks. We applied this correction in the revised MS.
- Line 32: List both transfer functions: logsig, tansig, - Maybe also the functions could be spelled out – log sig is probably the logistic transfer function
Response: Thanks. we listed them fully in the revised MS.
- Line 79-82: This is too vague, please be more specific
Response: Thanks for suggestion. We rephrased this sentence by making this more specific one. Now it is better understandable.
- Line 122: Sample plots or trees samples? If it is plots please give the number of trees.
Response: We modified the data materials sub-section substantially. Now everything is clearly understandable.
- Line 124-125: 20 and 96 does not sum up to 112 are these trees
Response: Thanks. We made mistake here. There are 112 sample plots which contain 20 sample plots data records in 1997 and 92 sample plots data records in 2002. We modified the data materials sub-section substantially. Now everything is clearly understandable.
- Line 169: Please give the reference for the formula
Response: Thanks. We cited a reference here, and rephrased a sentence in better way.
- Line 173, 174: The reference to equation 3 and 4 is missing, logsig – is this the logistic function, which is given in equation 4?
Response: Thanks. The logistic sigmoid function expression is as shown in Equation 2 and the tangent sigmoid function expression shown in Equation 3. We modified the texts here accordingly.
- Line 189-190: strictly speaking mapminmax scales inputs so it always falls in the interval [-1,1]
Response: Thanks. We agreed, and made slight correction here.
- Line 229: What was k in your study?
Response: Thanks, in our study, k=5, and it was clarified in the revised MS.
- Line 230-249: It might not be necessary to describe the k-fold cross-validation algorithm. Also I think there is a problem with the indices in this section: Line 236: Model Mj; Line 242: What does the index p stand for?
Response: Thanks for this suggestion. However, we decided to retain the description of the k-fold cross-validation algorithm as it is by considering the important roles of the k-fold cross-validation algorithm for selecting the optimal neural network height- diameter model. Mj stands for arbitrary model, and index p stands for a model number.
- Line 274-276: This is not clear to me
Response: Thanks. We rephrased the texts and made much clearer. Our message here is that difference of the average MSE of double-hidden layers from that of the triple-hidden layers was not substantially big, even though the double-hidden layer neural network, the structure of which is 1:5:5:1, had the smallest MSE.
- Table 5: There seem to be some lines which are duplicated
Response: Thanks. It was corrected in the revised MS.
- Line 315: In the first equation the slope coefficient seems to be missing
Response: Thanks. It was corrected in the revised MS.
- Line 316: I do not understand this sentence
Response: Thanks. We wanted to describe the relationship between the predicted values and observed values. We meant that if fitted line is closer to the diagonal line is, fitting precision would be higher. This information is very obvious and not necessary to introduce here, and therefore, we deleted this sentence, as graph itself is very clear for this information.
- Line 320: Why did you choose -1.1 m and -2.2 m?
Response: Thanks. Any interval could be used for making the comparison. However, we considered these intervals as our data points have scattered mostly between these intervals. It was clarified in the revised MS.
- Line 338: I don’t think performance-price is the correct expression here.
Response: We agreed and changed this.
- Line 339: Do you think your results are generalizable for different sample sizes?
Response: We considered that our results could be used for generalization to different sample sizes. Our study provides a way to choose a relatively optimal model. With a large sample size, there may be the need of more numbers of iterations, but we can still choose a relatively better model.
- Line 363: Do such interfaces exist in standard statistical software?
Response: We wrote the codes according to our proposed method in MatLab. The standard statistical software, e.g., SAS, R, and MatLab, have not the interfaces. But we may provide the Dynamic Link Library file upon a request.
- Line 382: What do you mean by unified method?
Response: It means the same or similar methods as ours, which we have conserved is an efficiently-organized method. And for the method of determining the structure of the neural network, there are few complete and systematic studies, but in our study, we propose a method to select the optimal structure of the neural network. It was clarified in the revised MS.
Typos
- Line 4: propagation
Response: Done.
21.Line 28: Populus spp. L.
Response: Done.
- Line 37: neural
Response: Done.
- Line 47: equations
Response: Done.
- Line 48: relatively
Response: Done.
- Line 52: the optimal
Response: Done.
- Line 54: the main theme
Response: Done.
- Line 59: site quality
Response: Done.
- Line 65: constituents
Response: Done.
- Line 83: Maybe better: giving precise results
Response: Done.
- …Please send out for English proof reading
Response: Thanks for this suggestion. We carefully checked the English language and edited wherever necessary. Now it is substantially better in terms of English grammar and manuscript quality as whole.
Reviewer 4 Report
This manuscript presents a method for training a Back Propagation Neural Network for modeling height-diameter relationships in Poplar plantations in China. While the work is interesting, there are issues with the manuscript that should be fixed before publishing. Largely, there are many grammar and spelling mistakes, most of which would have been caught with standard spell checker. For example, even the title has spelling errors where “Propagrtation” should be “Propagation”. These mistakes must be fixed and should have been addressed before going to review.
Overall, the application of a machine learning algorithm for modeling height-diameter relationships is quite interesting. Also, the comparison to existing parametric equations shows how machine learning algorithms can prove superior to the assumed relationships of non-linear models. What is missing is a discussion on model application. However, it is not clear if the manuscript is meant solely for the purpose of presenting a method or if the new model will actually be used for predicting height from diameter in poplar plantations in China. Regardless, some discussion on model application could be relevant. For example, the use of traditional models has the benefit of reasonable extrapolation when predicting outside of the range of the data used for model fitting. Machine learning algorithms are notorious for producing large errors when extrapolated. Some discussion on this and links to other publications could add to the manuscript.
General Comments
Don’t forget to cite the software! How were the models fitted and what packages were used? There are many references to least squares regression, however, the traditional models would be fit with non-linear least squares in their presented forms.
A plot of the data in table (1) relating height to diameter can help the reader understand the variation in the data. This can go in the supplementary information.
Lines 51-51: Can the authors provide some examples to solidify the point.
Line 62 (equation 1). This isn’t a general form of an equation but simply says that y is a function of multiple independent variables. This doesn’t add to the manuscript and I suggest removing it. A simple paragraph explain which independent variables are correlated with tree height would be sufficient.
Line 73: I would say that in more recently, with the advance of computing power and statistical fitting methods height-dbh models are fitted with non-linear least squares or non-linear mixed models. The authors have already made the case that height-diameter models are non-linear. Also, the second author has a manuscript from Norway on mixed-effects height diameter models. I would expect some small discussion on the updated methods.
Lines 140-142: I don’t agree with this statement. All of the equations in Table 2 are standard functions which are easily fitted and validated using traditional statistical techniques.
Spelling and Grammar (not an exhaustive list)
Line 4: change “Propagrtation” to “Propagation” , change “Plantation” to “Plantations”
Line 16: change «diammeter» to «diameter»
Line 31: change «structre» to «structure»
Line 59: change «qulity» to «quality»
Line 65: “contituents” to “constituents”?
Line 67: “desnsity” to “density”
Line 70: “addion” to “addition”
Line 71: “type” to “types”
Line 73: remove “the” before “least squares regression”
Line 123: change “plants” to “trees”
Line 146: change “neural network has” to “neural networks have”
Line 151: add “The” before “Main”
Line 368: change “parametrs” to “parameters”
Line 400: change “adeqauetly” to “adequately”
Author Response
This manuscript presents a method for training a Back Propagation Neural Network for modeling height-diameter relationships in Poplar plantations in China. While the work is interesting, there are issues with the manuscript that should be fixed before publishing. Largely, there are many grammar and spelling mistakes, most of which would have been caught with standard spell checker. For example, even the title has spelling errors where “Propagrtation” should be “Propagation”. These mistakes must be fixed and should have been addressed before going to review.
Overall, the application of a machine learning algorithm for modeling height-diameter relationships is quite interesting. Also, the comparison to existing parametric equations shows how machine learning algorithms can prove superior to the assumed relationships of non-linear models. What is missing is a discussion on model application. However, it is not clear if the manuscript is meant solely for the purpose of presenting a method or if the new model will actually be used for predicting height from diameter in poplar plantations in China. Regardless, some discussion on model application could be relevant. For example, the use of traditional models has the benefit of reasonable extrapolation when predicting outside of the range of the data used for model fitting. Machine learning algorithms are notorious for producing large errors when extrapolated. Some discussion on this and links to other publications could add to the manuscript.
Response:Thank you very much for this suggestion. Our main intention is that this article should provide the insights about the methods of choosing the best neural networks structure for an optimal H-D model and slight comparison of this model against H-D model resulted from traditional modeling approach. However, considering your and the fourth reviewer’s suggestions, we also fitted the mixed-effects model and compared its results against those from artificial neural network approach. We have added a new sub-section “3.4 Comparison to the mixed effects model accuracy” in the revised manuscript. We also briefly described the mixed-effect modeling in sub-section 2.2.2 as well. We found that fit statistics of the mixed effects H-D model obviously inferior to the BP neural network H-D model, but substantially superior to the height-diameter model fitted using ordinary least square regression. These contents were added in the discussion section in the revised MS. In addition, we also carefully checked the English language and edited wherever necessary. Now it is substantially better in terms of English grammar and manuscript quality as a whole.
General Comments
- Don’t forget to cite the software! How were the models fitted and what packages were used? There are many references to least squares regression, however, the traditional models would be fit with non-linear least squares in their presented forms.
Response:Thanks. The computations including the developed BP neural network height-diameter models and 5-fold cross-validation were implemented in the MatLab 2016b software. The developed mixed-effects height-diameter model were estimated by maximum likelihood using the Lindstrom and Bates (LB) algorithm implemented in the R software (version 3.2.2) nlme function. The traditional height-diameter functions using ordinary least square regression were implemented by the nls function in R software (version 3.2.2). These were clarified in the revised MS.
- A plot of the data in table (1) relating height to diameter can help the reader understand the variation in the data. This can go in the supplementary information.
Response:Thanks. We presented them in supplementary file in annex in the revised MS.
- Lines 51-51: Can the authors provide some examples to solidify the point.
Response:Thanks, some typical references were cited in the revised MS.
- Line 62 (equation 1). This isn’t a general form of an equation but simply says that y is a function of multiple independent variables. This doesn’t add to the manuscript and I suggest removing it. A simple paragraph explain which independent variables are correlated with tree height would be sufficient.
Response: We agreed, and deleted this Equation 1. Now it seems better as text presented there provides sufficient information that height is strongly correlated to a multiple factors.
- Line 73: I would say that in more recently, with the advance of computing power and statistical fitting methods height-dbh models are fitted with non-linear least squares or non-linear mixed models. The authors have already made the case that height-diameter models are non-linear. Also, the second author has a manuscript from Norway on mixed-effects height diameter models. I would expect some small discussion on the updated methods.
Response:Thanks for suggestion. In the beginning, we were focusing on the developing of the height-diameter models using some artificial neural networks approaches and comparison of their results against the traditional height-diameter modeling approach which we have defined as least squared regression approach. Our intention is that this article should provide the insights about the methods of choosing the best neural networks structure for optimal height-diameter model. There would be better comparison between the traditional modeling and neural network modeling when the same numbers of input variables are used for training the models. When mixed-effects modeling is considered for comparison, it need to have more input variables than those of traditional and neural networks modeling, as mixed-effects model is always associated with additional random components. However, considering the suggestion provided from 4th reviewer, we also carried out the comparison of the neural networks modeling, traditional modeling and mixed-effects modeling and presented the results in the revised manuscript. Here in the introduction, we still considered not writing a large bulk of the texts about the mixed-effects modeling. However, as per your suggestion, we have added some relevant texts about mixed-effects models with some citations including reference you pointed out here. More descriptions about mixed-effects models were given in the sections of the methodology, results, and discussion.
- Lines 140-142: I don’t agree with this statement. All of the equations in Table 2 are standard functions which are easily fitted and validated using traditional statistical techniques.
Response:We agreed and modified the texts accordingly.
- Spelling and Grammar (not an exhaustive list)
Line 4: change “Propagrtation” to “Propagation”, change “Plantation” to “Plantations”
Response: Done.
- Line 16: change «diammeter» to «diameter»
Response: Done.
- Line 31: change «structre» to «structure»
Response: Done.
- Line 59: change «qulity» to «quality»
Response: Done.
- Line 65: “contituents” to “constituents”?
Response: Done.
- Line 67: “desnsity” to “density”
Response: Done.
- Line 70: “addion” to “addition”
Response: Done.
- Line 71: “type” to “types”
Response: Done.
- Line 73: remove “the” before “least squares regression”
Response: Done.
- Line 123: change “plants” to “trees”
Response: Done.
- Line 146: change “neural network has” to “neural networks have”
Response: Done.
- Line 151: add “The” before “Main”
Response: Done.
- Line 368: change “parametrs” to “parameters”
Response: Done.
- Line 400: change “adeqauetly” to “adequately”
Response: Done.
Round 2
Reviewer 2 Report
The manuscript quality has been significantly improved. The inclusion of mixed effects model was a good idea and make a better view to whole manuscript. I found only few mistakes in the reference list again. After these minor corrections I recommend accept the manuscript.
Line 520: Replace Dieguezaranda with Dieguez Aranda.
Line 536: Replace Fernándezmartínez with Fernández Martínez.
Line 545: Replace Crecentecampo with Crecente Campo.
Author Response
The manuscript quality has been significantly improved. The inclusion of mixed effects model was a good idea and make a better view to whole manuscript. I found only few mistakes in the reference list again. After these minor corrections I recommend accept the manuscript.
Response: Thanks for your positive comments on our manuscript (MS). The comments and suggestions that you gave are all valuable and helpful for revising and improving our MS. We have read the comments and suggestions carefully and made the corresponding corrections and clarifications.
1. Line 520: Replace Dieguezaranda with Dieguez Aranda.
Response: It was corrected in the revised MS.
2. Line 536: Replace Fernándezmartínez with Fernández Martínez.
Response: It was corrected in the revised MS.
3. Line 545: Replace Crecentecampo with Crecente Campo.
Response: It was corrected in the revised MS.